# Heavy-Tailed Class Imbalance and Why Adam Outperforms Gradient Descent on Language Models

**Frederik Kunstner**[1]
kunstner@cs.ubc.ca

**Alan Milligan**[1]
alanmil@cs.ubc.ca

**Robin Yadav**[1]
robiny12@student.ubc.ca

**Mark Schmidt**[1,2]
schmidtm@cs.ubc.ca

**Alberto Bietti**[3]
abietti@flatironinstitute.org

[1] University of British Columbia      [2] Canada CIFAR AI Chair      [3] Flatiron Institute

## Abstract

Adam has been shown to outperform gradient descent on large language models by a larger margin than on other tasks, but it is unclear why. We show that a key factor in this performance gap is the heavy-tailed class imbalance found in language tasks. When trained with gradient descent, the loss of infrequent words decreases more slowly than the loss of frequent ones. This leads to a slow decrease on the average loss as most samples come from infrequent words. On the other hand, Adam and sign-based methods are less sensitive to this problem. To establish that this behavior is caused by class imbalance, we show empirically that it can be reproduced across architectures and data types, on language transformers, vision CNNs, and linear models. On a linear model with cross-entropy loss, we show that class imbalance leads to imbalanced, correlated gradients and Hessians that have been hypothesized to benefit Adam. We also prove that, in continuous time, gradient descent converges slowly on low-frequency classes while sign descent does not.

## 1 Introduction

The recent success of large language models such as GPT-3 (Brown et al., 2020) and its successors has relied on costly training procedures at unprecedented scale. A key ingredient in their training is the Adam optimizer (Kingma and Ba, 2015), which outperforms stochastic gradient descent (SGD) on language problems by a large margin. Despite this large performance gap, we have a poor understanding of why Adam works better and it has been difficult to find new optimizers that consistently improve over Adam (Schmidt et al., 2021). Not only is it computationally difficult to validate new optimizers on large models, but we also lack theoretical guidance; we do not know what "problem" Adam solves to outperform SGD.

The success of Adam on language transformers has been well documented. Multiple works have found metrics or statistics that correlate with the improved performance of Adam, showing that it yields uniform updates across parameters despite imbalanced gradients (Liu et al., 2020), gives a better descent direction than the gradient (Pan and Li, 2023), and takes a path over which a robust variant of the condition number is smaller (Jiang et al., 2022). But these observations do not provide a mechanism explaining what property of the problem leads to the improved performance of Adam.

Plausible mechanisms have been put forward, but they do not provide a complete explanation. Zhang et al. (2020b) show that Adam-like methods are more resilient to heavy-tailed noise, which seems more prominent in language than in vision tasks. But noise is not the primary cause of the gap, as it already appears in deterministic training (Kunstner et al., 2023). An alternative hypothesis is that the magnitude of the gradient and Hessian are correlated, which justifies clipping (Zhang et al., 2020a). But to justify methods that normalize element-wise, like Adam and sign-like methods, we additionally need the gradient and Hessian to be correlated across parameters (Crawshaw et al., 2022). While there is empirical evidence for this behavior in neural networks, we do not have a good understanding of why this occurs, nor why this would be more pronounced on language rather than vision tasks.

38th Conference on Neural Information Processing Systems (NeurIPS 2024).

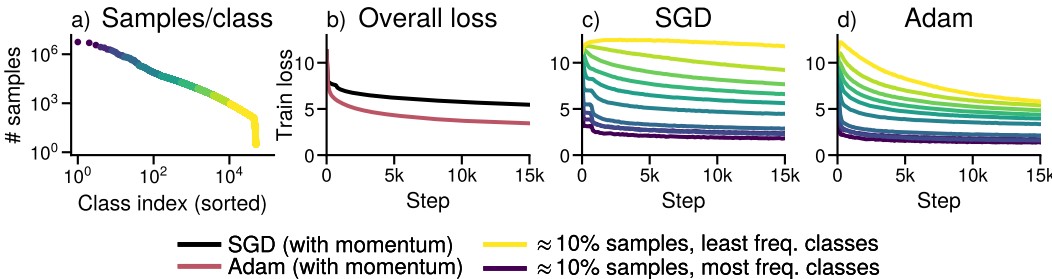

Figure 1: **Gradient descent does not make progress on low-frequency classes, while Adam does.** Training GPT2-Small on WikiText-103. **(a)** Distribution of the classes sorted by class frequency, split into groups corresponding to $\approx 10\%$ of the data. **(b)** Overall training loss. **(c, d)** Training loss for each group using SGD and Adam. SGD makes little to no progress on low-frequency classes while Adam makes progress on all groups. **(b)** is the average of **(c, d)** for the respective optimizer.

## 1.1 Contributions

Our goal is to answer the following question: *what is the "problem" that makes SGD slow on language tasks, that Adam "fixes" to perform better?*

**We argue the problem is what we call heavy-tailed class imbalance,** where rare classes account for a large fraction of the data. Language data is imbalanced as some words are much more frequent than others, typically following a power-law. A common modeling assumption is Zipf's law, where the $k$th most frequent word has frequency $\propto 1/k$ (Piantadosi, 2014). For language tasks framed as next-token prediction, this property is reflected in the tokens and leads to heavy-tailed class imbalance. This contrasts with typical vision datasets such as MNIST, CIFAR, and ImageNet, which are curated to have uniform classes, but also with imbalanced problems with a small number of classes. For example, in binary classification, extreme imbalance implies the minority class has a limited impact on the loss; with an imbalance of 99:1, only 1% of the data comes from the minority class.

**The performance gap arises because SGD makes slow progress on rare classes, see Figure 1.** On a binary problem, slow performance on 1% of the data need not have a large impact on the average loss if we make fast progress on the remaining 99% of the samples. In contrast, the heavy-tailed class imbalance found in language tasks makes it possible for low-frequency classes to account for most of the data and significantly contribute to the loss, leading to slow performance overall.

**We show that heavy-tailed class imbalance makes SGD slow across tasks in Section 2.** We show that modifying vision datasets to exhibit heavy-tailed imbalance leads to slow progress with SGD on architectures where the performance gap with Adam is typically smaller. The impact of heavy-tailed imbalance can even be seen on linear models. Additionally, the performance of SGD improves with techniques that address imbalance such as upweighting rare classes.

**Our findings provide a simple model where Adam outperforms SGD,** a softmax linear model under heavy-tailed class imbalance, which we analyze in Section 3. We show empirically that a correlation between the magnitude of the gradient and Hessian across coordinates, used to justify the benefits of Adam, appears naturally even on a linear model with class imbalance. We provide intuition as to how this pattern emerges through an assignment mechanism that leads to a correlation between class frequencies and the magnitude of the gradient and Hessian across parameters. We additionally prove that, on a simple dataset and in continuous time, GD is slow on low-frequency classes while sign descent is insensitive to the class frequencies.

We do not claim that class imbalance is the only reason Adam outperforms SGD, as other properties of the data or architectures likely also contribute to this gap. Instead, we show that Adam consistently outperforms SGD under heavy-tailed class imbalance. The difficulty of minimizing the loss of minority classes has been explored for binary problems or problems few classes (Anand et al., 1993; Francazi et al., 2023), but the recent scaling of large language models to predictions over more than 100 000 classes puts the problem on a new scale. Our findings indicate that heavy-tailed class imbalance has a significant impact on training performance and should be a consideration for future optimizers to perform well on language and other tasks exhibiting heavy-tailed class imbalance.

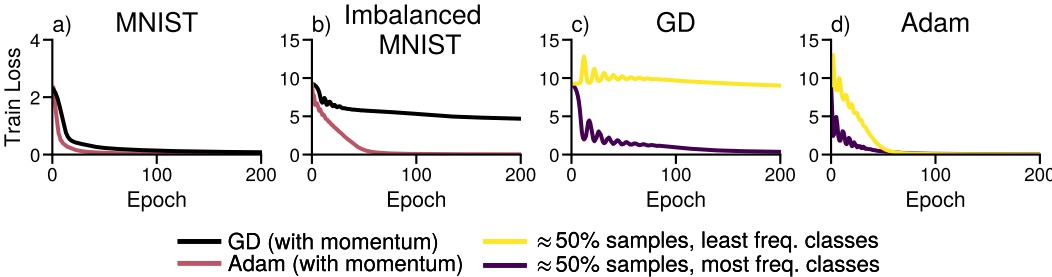

Figure 2: **Adam outperforms GD for training a CNN under heavy-tailed class labels. (a)** Performance on the MNIST dataset. **(b)** Performance on a modified MNIST with two groups of classes. The first group consists of the 10 original classes with $\approx 5k$ samples each, while the second consists of $\approx 10k$ added classes with 5 examples each. **(c, d)** Performance of GD and Adam on the two groups. The initial loss is higher for imbalanced MNIST as there are $\approx 10^4$ classes instead of 10, leading to a loss of $-\log(1/10^4) \approx 9.2$ for a uniform prediction instead of $-\log(1/10) \approx 2.3$.

## 2 Experimental results and ablation studies

Figure 1 suggests a correlation between class frequencies and optimization performance that impacts SGD more than Adam. The goal of this section is to verify that (i) class imbalance is a root cause for the performance gap between SGD and Adam, and (ii) whether this gap can be reproduced with simpler algorithms, such as deterministic optimizers, or using sign descent as a proxy for Adam.

To test these hypotheses, we perform experiments focusing on the training loss as our objective is to understand what makes optimization difficult. We use a simple training procedure, with a constant step-size tuned by grid search. For visualization, we split the data into groups of classes with similar frequencies, as in Figure 1. For instance, for 10 groups, the first group corresponds to $\approx 10\%$ of the samples from the most frequent classes. This grouping is only used for visualization and does not affect training. The models, datasets and training procedures are described in Appendix A.

In Appendix B, we give additional information and additional ablation experiments on language models. We show that the heavy-tailed class distribution appears across datasets and tokenizers, and that the separation across class frequencies observed on the training loss in Figure 1 also affects the validation loss. We show that similar dynamics appear on smaller language models, including when training only the last layer while keeping the embedding and attention modules frozen at initialization. Finally, we show that stochasticity is not necessary to reproduce the impact of heavy-tailed class imbalance, and that it also appears when using deterministic updates (i.e., GD instead of SGD). As a result, we use deterministic updates whenever possible, denoted by GD in the figures.

### 2.1 Reproducing the frequency gap with vision models

Language transformers are often contrasted with vision CNNs, where we do not see a large performance gap between SGD and Adam. Our hypothesis is that a key differentiation between the two settings is the heavy-tailed class imbalance present in language data. In this section, we show that making heavy-tailed vision datasets leads to slower performance with SGD and a larger performance gap with Adam. These experiments show that heavy-tailed imbalance has a significant impact on performance and can make an otherwise "easy" problem into a "hard" one for SGD.

**CNN.** We first use a CNN on a variant of MNIST with heavy-tailed class imbalance. We augment the dataset to have two equally-sized groups of classes with a relative frequency difference of 1000. The first group consists of the original 10 classes with $\approx 5k$ samples/class. For the second, we create $\approx 10k$ new classes with 5 samples/class. We create new classes by copying existing images and adding a "barcode" in a corner of the image, see Appendix A. The performance of GD and Adam is shown in Figure 2. On the original MNIST dataset, both optimizers drive the loss to 0, and Adam still makes progress on both groups in the imbalanced case. But on the imbalanced variant, GD makes almost no progress on half of the data corresponding to the low-frequency classes and progress stalls. However, it eventually converge if run for much longer (see Appendix D.2), indicating that the problem is one of slow optimization rather than getting stuck in a local minima.

**ResNet.** We replicate this effect with a ResNet18 on an imbalanced variant of ImageNet. We subsample classes with frequencies $\pi_k \propto 1/k$ and compare against a uniform subset with a similar

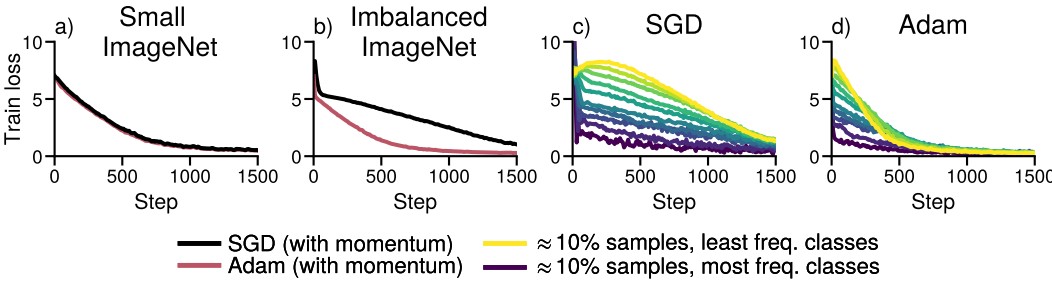

Figure 3: **Adam outperforms SGD for training a ResNet under heavy-tailed class labels. (a)** Performance on a subset of ImageNet and **(b)** an imbalanced subset of ImageNet with class frequencies $\pi_k \propto 1/k$. **(c, d)** Performance of GD and Adam on groups corresponding to $\approx$10% of the data.

number of samples. In Figure 3, we see that SGD and Adam perform similarly on uniform data but a performance gap appears across class frequencies on the heavy-tailed imbalanced dataset. As in Figures 1 and 2, SGD is slower on imbalanced data, especially on low-frequency classes.

**Vision Transformers.** This performance gap also appears with vision transformers (ViTs). In Appendix C, we see that SGD and Adam both perform well on ImageNet, but exhibit a similar performance gap as in Figure 1 on the imbalanced variant. While ViTs may require more raw data, data augmentations, or regularization to generalize as well as ResNets (Steiner et al., 2022), there does not seem to be a large gap between SGD and Adam without class imbalance.

### 2.2 Reproducing the frequency gap with a linear model on uniform data

To highlight that heavy-tailed imbalance alone can lead to the observed difficulties, we reproduce this behavior in a simple setting: a softmax linear model with cross-entropy loss. We create a dataset where the class frequencies approximate $\pi_k \propto 1/k$ and draw $n$ samples uniformly from $[0, 1]$ in $d$ dimensions, independently of the label. While there is no relationship to learn, the optimization problem is still well posed and a linear model can separate the data if $n \ll d$. As on the transformer of Figure 1, GD makes less progress on low-frequency classes than Adam, as shown in Figure 4.

This example illustrates that a problem that might look innocuous at first is hard to optimize with GD due to heavy-tailed imbalance, while the performance of Adam is less negatively impacted. Nonetheless, imbalance alone is not sufficient to make GD slow. It is possible to generate pathological datasets with heavy-tailed imbalance where GD fits all classes fast, by making all the samples (close to) orthogonal. In this case, each sample is learned independently of the others, and there is no difference across classes. However, perfectly orthogonal data is unlikely, especially as we expect samples from similar classes to be assigned a similar (correlated) representation. We discuss this issue and give additional examples on the linear model in Appendix D.

### 2.3 Interactions between optimizer and imbalance

We have shown that heavy-tailed class imbalance can lead to different performance across class frequencies, but it is not clear which component of the training process has the highest impact on this behavior. We next experiment with simple algorithms to answer the following questions. (i) Is the impact of class imbalance due to stochasticity, or does it happen with deterministic training? (ii) Which component of Adam leads to an improved performance? and (iii) If imbalance is the problem, can we improve the performance of SGD by reweighting the losses?

**Class imbalance already impacts deterministic optimization.** A natural hypothesis to explain the impact of class imbalance is that it may be due to small batch sizes in SGD; rare classes could be sampled less often, and thus learned more slowly. On the other hand, stochasticity has been found to have little impact on the gap between SGD and Adam (Kunstner et al., 2023). Our experiments in Figures 2, 4 and 5 and further examples in Appendix B.3 reproduce the dynamics of Figure 1 with full batch GD and Adam, indicating the problem already arises in the deterministic setting.

**Adam and sign descent both perform well under imbalance.** Following Kunstner et al. (2023), we check whether the benefit of Adam is due to a change in the magnitude of the update or its direction. Changing the magnitude as in normalized GD is known to perform better on separable problems (Nacson et al., 2019), while the benefits of Adam have been attributed to the change of direction close

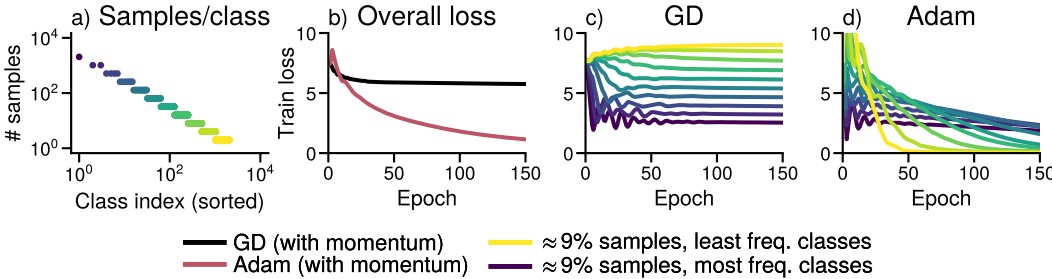

Figure 4: **The impact of heavy-tailed class imbalance is reproducible with linear models.** Softmax regression on synthetic data. The inputs are drawn from a uniform distribution on $[0,1]^d$. The target classes are heavy-tailed **(a)** and independent of the inputs, but the model can still fit the data as it is overparameterized. **(b, c, d)** Overall training loss and performance of GD and Adam on each subset.

to sign descent (Tieleman and Hinton, 2012; Balles and Hennig, 2018). We compare the performance of GD, Adam, normalized GD and sign descent, with and without momentum, for training the last layer of a small transformer in Figure 5 and on additional problems in Appendix E. Normalization and momentum helps across problems, but they have less impact on the performance gap across class frequencies than changing the update direction. Sign descent and Adam have a similar performance.

**Upweighting low-frequency classes can help.** Given our hypothesis that the performance gap between (S)GD and Adam is due to class imbalance, we expect interventions directly targeting imbalance to improve performance. In Appendix E.1, we show that upweighting the loss of low-frequency classes can improve the performance of SGD. While reweighting is not complete solution as it changes the objective function, this experiment supports the hypothesis that the optimization problem is due to heavy-tailed class imbalance.

## 3 An investigation on linear models

Heavy-tailed imbalance already leads to slow performance on the linear softmax model of Figure 4, but we do not have a good understanding of why GD becomes slow while Adam is less affected. In this section, we explore the effect of heavy-tailed class imbalance on the special case of softmax linear models, showing that it leads to correlated, imbalanced gradients and Hessians. In Section 3.1, we give an example on a quadratic where imbalanced Hessians lead to a performance gap between GD and Adam. In Section 3.2, we show that class imbalance leads to imbalanced gradients and Hessians that are correlated with class frequencies through an *assignment mechanism*, showing that this pattern emerges naturally. Finally, we prove that on a simple imbalanced problem and in continuous time, GD is slow on low-frequency classes while sign descent is fast on all classes in Section 3.3.

### 3.1 Intuition on a weighted quadratic problem

Consider the following toy problem which is purposefully oversimplified to provide a high-level intuition about the optimization dynamics. Suppose we have $c$ functions $f_1, ..., f_c$, corresponding to the losses for each class, that are on the same scale in the sense that gradient descent with step-size $\alpha$ makes fast progress on any $f_i$. For concreteness, take $f_i(w) = \frac{1}{2}\|w\|^2$, where GD with a step-size of 1 converges in one step. Instead of running GD on each function independently, suppose we run GD on the weighted average $f(w_1, ..., w_c) = \sum_{i=1}^{c} \pi_i f_i(w_i)$ with positive weights $\pi_1 \geq ... \geq \pi_c$, $\sum_i \pi_i = 1$, corresponding to the class frequencies. If these weights span multiple orders of magnitude, we expect a similar behavior as in Figures 1 to 5, as illustrated in Figure 6. GD makes slow progress on functions with low weights as the gradient w.r.t. $w_k$ is scaled by $\pi_k$,

$$w_k^{(t)} = w_k^{(t-1)} - \alpha \pi_k f_k'(w_k^{(t-1)}) = (1 - \alpha \pi_k)^t w_k^{(0)}.$$

This slow convergence on functions with low weights cannot be fixed by increasing the step-size, as increasing it beyond $1/\pi_1$ would cause instabilities on the highest-frequency "class" $f_1$. The problem is that we use the same step size for all functions, which have different scales. Adam and sign descent

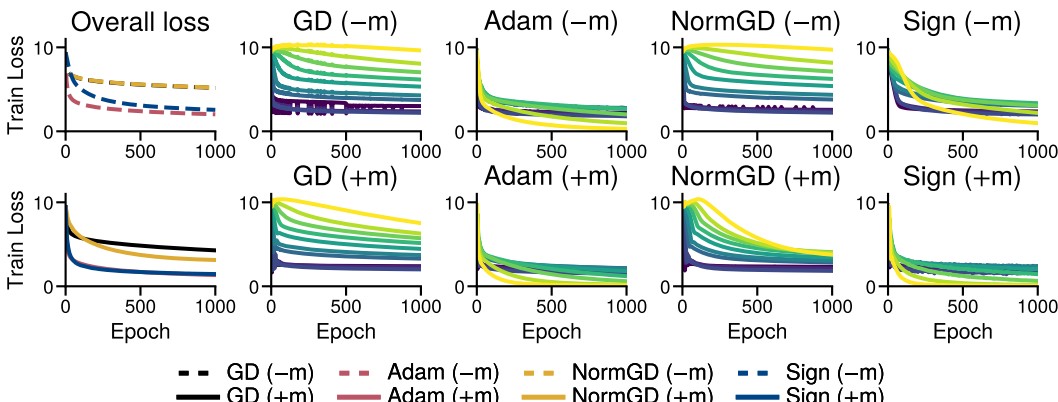

Figure 5: **Sign descent, as a simplified form of Adam, performs well on low-frequency classes.** Training the last layer of a simplified one-layer transformer with GD, Adam, normalized GD, and sign descent, with and without momentum ($\pm$m). Momentum and normalizing the magnitude help but have smaller effects than using sign descent, which recovers similar dynamics to Adam.

are less sensitive to this problem as their updates are independent of $\pi_k$,

$$w_k^{(t)} = w_k^{(t-1)} - \alpha \frac{\pi_k f_k'(w_k^{(t-1)})}{\left| \pi_k f_k'(w_k^{(t-1)}) \right|} = w_k^{(t-1)} - \alpha \operatorname{sign}(f_k'(w_k^{(t-1)})).$$

While sign descent or Adam with a fixed step-size need not converge and can oscillate around the minimum, they perform much better in early iterations, independently of $\pi_k$.

Another perspective is that the imbalance in the weights $\pi_1, ..., \pi_c$ makes the problem ill-conditioned. The weights not only affect the gradient of $f$ but also its Hessian, which is $\operatorname{Diag}([\pi_1, ..., \pi_c])$. A common intuition for Adam is that using the magnitude of the coordinates of the gradient as a preconditioner is a good proxy for the Hessian diagonal (Duchi et al., 2011; Kingma and Ba, 2015), which would also lead to larger step-sizes for coordinates with small $\pi_k$. While this does not hold in general (Kunstner et al., 2019), the gradient can be a reasonable approximation to the Hessian on this problem. The gradient is $[\pi_1 w_1, ..., \pi_c w_c]$. If the weights $\pi_1, ..., \pi_c$ vary by orders of magnitude more than the parameters $|w_1|, ..., |w_c|$, the gradient and Hessian will be correlated, and preconditioning by the gradient magnitude or Hessian diagonal will yield similar directions.

### 3.2 Correlations between the magnitude of the gradient and Hessian across coordinates

What is lacking to explain Adam's improved performance is an understanding of how a correlation between the gradient and Hessian arises in realistic problems. This feature has been observed on neural networks, but we do not yet know why it appears, even on the softmax linear problem. The caricature of the diagonal quadratic problem of the previous section provides some intuition, but does not directly apply to the softmax linear model of Figure 4 as that problem is neither quadratic nor separable. Nonetheless, a similar pattern emerges in the rows $\mathbf{w}_1, ..., \mathbf{w}_c$ of its parameter matrix $\mathbf{W} \in \mathbb{R}^{c \times d}$; the magnitude of the gradient and Hessian across rows and the class frequencies can become correlated during training due to class imbalance. In this section, we establish this observation empirically and provide a mechanism for how it emerges.

In Figure 7, we show the gradient norm against the Hessian trace with respect to each row $\mathbf{w}_k$ throughout the trajectory of Adam on the softmax linear model of Figure 4. While there is no correlation at initialization, the gradient and Hessian blocks become correlated with class frequencies during training and become imbalanced. This imbalance in the diagonal blocks is the main feature of the Hessian as the than off-diagonal blocks are orders of magnitude smaller, as shown in Figure 9. Similar dynamics occur with GD, although only on high-frequency classes as GD makes little progress on low-frequency classes, see Appendix F. This correlation also appears in the last layer of large models such as GPT2-Small used in Figure 1, as shown in Figure 8.

To explain this behavior, we show that the impact of samples on the Hessian follows an *assignment mechanism*: if the model assigns samples to their correct class, the Hessian with respect to $\mathbf{w}_k$ is

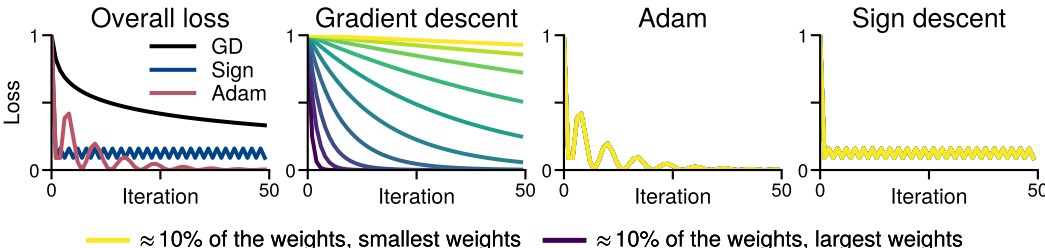

Figure 6: **Class-separation on the quadratic problem of Section 3.1 with weights $\pi_k \propto 1/k$.** GD fits functions with low weights more slowly, while Adam and sign descent have the same dynamics across all functions and all the lines overlap as every parameter $w_i$ is initialized at $w_i = 1$.

primarily influenced by samples from class $k$, leading to a correlation between the magnitude of the gradient, Hessian, and class frequencies. To capture this effect, we introduce some notation and a simplifying assumption. Suppose we have $n$ samples with inputs $\mathbf{x}_i \in \mathbb{R}^d$ and labels $y_i \in [c]$, where class $k$ has frequency $\pi_k = n_k/n$. The parameters of the linear model are $\mathbf{W} \in \mathbb{R}^{c \times d}$. We write $\mathbf{p}(\mathbf{x}) = \sigma(\mathbf{W}\mathbf{x})$ for the predicted probabilities where $\sigma$ is the softmax, and summarize the data as

$$\bar{\mathbf{x}} = \tfrac{1}{n}\sum_{i=1}^n \mathbf{x}_i, \quad \bar{\mathbf{x}}^k = \tfrac{1}{n_k}\sum_{i:y_i=k}\mathbf{x}_i, \quad \bar{\mathbf{H}} = \tfrac{1}{n}\sum_{i=1}^n \mathbf{x}_i\mathbf{x}_i^\top, \quad \bar{\mathbf{H}}^k = \tfrac{1}{n_k}\sum_{i:y_i=k}\mathbf{x}_i\mathbf{x}_i^\top.$$

**Assumption 1** (correct assignment). The model correctly assigns samples to class $k$ if it predicts $k$ with non-negligible probability $p$ on samples from that class ($(\mathbf{p}(\mathbf{x}_i))_k = p = \omega(1/c)$ for $\mathbf{x}_i$ from class $y_i = k$), and predicts $k$ with near-random chance otherwise ($(\mathbf{p}(\mathbf{x}_i))_k = O(1/c)$ for $\mathbf{x}_i$ where $y_i \neq k$).

**Proposition 2.** *If initialized at $\mathbf{W}_0 = 0$, the gradient and Hessian of the loss $\mathcal{L}$ w.r.t. $\mathbf{w}_k$ are*

$$\nabla_{\mathbf{w}_k}\mathcal{L}(\mathbf{W}_0) = \pi_k\bar{\mathbf{x}}^k - \tfrac{1}{c}\bar{\mathbf{x}}, \qquad\qquad \nabla^2_{\mathbf{w}_k}\mathcal{L}(\mathbf{W}_0) = \tfrac{1}{c}\big(1 - \tfrac{1}{c}\big)\bar{\mathbf{H}}, \qquad (1)$$

*During training, if the model correctly assigns samples to class $k$ with probability $p$ (Assumption 1),*

$$\begin{aligned}\nabla_{\mathbf{w}_k}\mathcal{L} &= (1-p)\pi_k\,\bar{\mathbf{x}}^k + O\big(\tfrac{1}{c}\big),\\ \nabla^2_{\mathbf{w}_k}\mathcal{L} &= p(1-p)\pi_k\,\bar{\mathbf{H}}^k + O\big(\tfrac{1}{c}\big),\end{aligned} \quad \text{and} \quad \|\nabla_{\mathbf{w}_k}\mathcal{L}\| \sim \left(\frac{1}{p}\frac{\|\bar{\mathbf{x}}^k\|}{\mathrm{Tr}(\bar{\mathbf{H}}^k)}\right)\mathrm{Tr}(\nabla^2_{\mathbf{w}_k}\mathcal{L}) \ \text{as } c \to \infty, \quad (2)$$

*for classes where the frequency does not vanish too quickly, $\pi_k = \omega(1/c)$.*

The assumption that $c \to \infty$ is used to obtain a simple and interpretable equation in the correlation. In practice, $c > 10^3$ appears sufficient to make the dependence on $\pi_k$ appear, as in Figures 7 and 8.

At initialization, Equation (1) shows that the Hessian blocks are uniform across classes while the gradients depend on $\pi_k$. If the data is uniform across classes ($\|\bar{\mathbf{x}}^k\| \approx \|\bar{\mathbf{x}}^{k'}\|$) while the frequencies differ by orders of magnitude, the the gradient blocks will mirror the class frequencies for high-frequency classes where $\pi_k \gg 1/c$. This confirms the pattern observed at initialization in Figures 7 and 8. During training, Equation (2) indicates a correlation between gradient norm and Hessian trace if classes have similar values of $\|\bar{\mathbf{x}}^k\|$, $\mathrm{Tr}(\bar{\mathbf{H}}^k)$ and predicted probabilities $p$, confirming the behavior observed during training in Figures 7 and 8 for the high frequency classes. As Adam fits low-frequency classes faster in Figure 4, they have a value of $p$ closer to 1 (shown in Appendix F) and deviate slightly from the trend in Figure 7, as expected from Equation (2).

We now give the main intuition and defer the derivation of the asymptotics to Appendix G. We ignore off-diagonal blocks here, as they are orders of magnitude smaller than diagonal blocks (Figure 9), and show in Appendix G.1 that they are expected to be small.

*Proof idea.* Our loss is $\mathcal{L}(\mathbf{W}) = \frac{1}{n}\sum_{i=1}^n \ell(\mathbf{W}, \mathbf{x}_i, \mathbf{y}_i)$, where $\ell$ is a softmax linear model,

$$\ell(\mathbf{W}, \mathbf{x}, y) = -\log(\sigma(\mathbf{W}\mathbf{x})_y), \quad \text{with} \quad \sigma(\mathbf{z})_k = \frac{\exp(\mathbf{z}_k)}{\sum_j \exp(\mathbf{z}_j)}. \qquad (3)$$

Writing $\mathbf{p}(\mathbf{x}) = \sigma(\mathbf{W}\mathbf{x})$ for the vector predicted probabilities, the gradient and Hessian blocks are

$$\nabla_{\mathbf{w}_k}\ell(\mathbf{W}, \mathbf{x}, y) = (\mathbf{1}[y = k] - \mathbf{p}(\mathbf{x})_k)\mathbf{x}, \qquad \nabla^2_{\mathbf{w}_k}\ell(\mathbf{W}, \mathbf{x}, y) = \mathbf{p}(\mathbf{x})_k(1 - \mathbf{p}(\mathbf{x})_k)\mathbf{x}\mathbf{x}^\top. \qquad (4)$$

The contribution of a sample $(\mathbf{x}, y)$ to the gradient w.r.t. $\mathbf{w}_k$ primarily depends on whether the sample belongs to class $k$ through the $\mathbf{1}[y = k]$ term, while the contribution to the Hessian block depends on whether the model assigns that sample to class $k$ through $\mathbf{p}(\mathbf{x})_k$. At initialization, $\mathbf{p}(\mathbf{x})_k = 1/c$

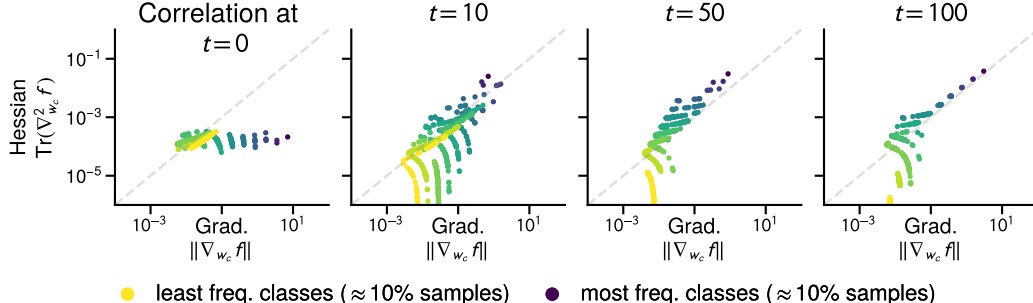

Figure 7: **The gradient norm and Hessian trace across blocks become correlated during training,** over the path taken by Adam in training the linear model of Figure 4. The blocks correspond to the rows $\mathbf{w}_1, ..., \mathbf{w}_c$ of the parameter matrix $\mathbf{W}$. The color indicates the class frequency, showing that lower (higher) frequency classes have smaller (larger) gradient norm and Hessian trace.

for all samples, and averaging the terms in Equation (4) yields Equation (1). Highlighting this effect during training is more challenging due to the dependency on the predictions. However, if $\mathbf{W}$ start to assign samples to their correct classes (Assumption 1), we can obtain a similar decomposition as Equation (1). For a given class $k$, the probabilities for correct labels are all $p$ while the probabilities for incorrect ones are bounded by $O(1/c)$, which vanishes in the limit of $c \to \infty$. □

This assignment mechanism explains why the gradient, Hessian, and class probabilities can become correlated on the linear model. While the gradient does not directly approximate the Hessian, the main feature of the imbalance in the Hessian comes from the weighting by the class frequencies $\pi_1, ..., \pi_c$, which is present in both the gradient and the Hessian, as shown in Figures 7 and 9. This correlation is not a global property of the problem, as there are parameters for which the opposite pattern holds, see Appendix F, but it appears during training if the optimization algorithm makes progress. While the per-coordinate normalization of Adam or sign descent was not designed to specifically address class imbalance, they appear to benefit from this property to make faster progress.

Our results complement prior work on optimization with class imbalance on problems with two or few classes, which argued that the gradient is dominated by the majority class, and as a result is biased towards making progress on the majority class at the expense of the minority class (Anand et al., 1993; Ye et al., 2021; Francazi et al., 2023). While this explains why GD might not make fast progress on rare classes, it was not clear why this would lead to slow performance on average, especially under heavy-tailed imbalance where there is no "majority". Our results show that, in addition to imbalance in the gradients, class imbalance leads to optimization difficulties through imbalanced Hessians.

### 3.3 Improvement of sign-based approaches over gradient descent

While the above arguments provide a high-level intuition as to why the gradient might be a reasonable proxy for the Hessian, it remains difficult to formally describe this effect and prove the benefits of Adam over GD without strong assumptions. Doing so would require a fine-grained analysis of the dynamics, as the correlation only appears during training. To obtain a provable a guarantee highlighting the benefit of sign-based methods, we consider a stripped-down problem where the only difficulty lies in the class imbalance:

**Simple imbalanced setting.** *Consider $c$ classes with frequencies $\pi_1, ..., \pi_c$ where all samples from a class are the same, $\mathbf{x}_i = \mathbf{e}_k$ if $y_i = k$, where $\mathbf{e}_k$ is the $k$th standard basis vector in $\mathbb{R}^c$.*

This setting is trivial as a possible solution is $\mathbf{W} = \alpha\mathbf{I}$ with $\alpha \to \infty$, or taking one step of gradient descent with an arbitrarily large step-size. However, we will see that the dynamics with small step-sizes already exhibit the separation by class frequencies observed experimentally. In this simplified setting, we show that the continuous time variant of gradient descent, gradient flow, and sign descent as a proxy for Adam, obtain qualitatively different convergence rates (proof in Appendix H).

**Theorem 3.** *On the* simple imbalanced setting, *gradient flow and continuous time sign descent initialized at $\mathbf{W} = 0$ minimize the loss of class $k$, $\ell_k(t) = -\log(\sigma(\mathbf{W}(t)\mathbf{e}_k)_k)$, at the rate*

$$\textit{Gradient flow:} \quad \ell_k(t) = \Theta(1/\pi_k t), \quad \textit{Continuous time sign descent:} \quad \ell_k(t) = \Theta(e^{-ct}).$$

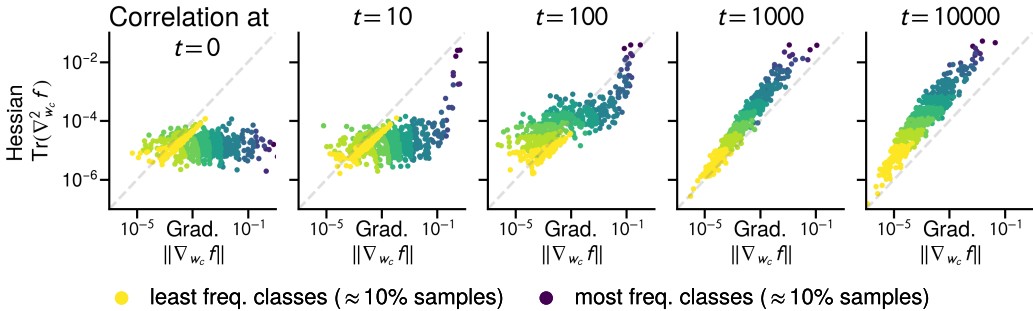

Figure 8: **The gradient-Hessian blocks also become correlated in the last layer of large models.** Reproducing Figure 7 on the transformer of Figure 1. Evolution of the gradient norm and Hessian trace for each row $\mathbf{w}_c$ of the last layer through optimization with Adam. Colors indicates class frequency. Lower (higher) frequency classes have smaller (larger) gradient norm and Hessian trace.

The difference between the sublinear rate of gradient flow ($1/t$) and linear rate of sign descent ($e^{-t}$) is similar to existing results for separable logistic regression, where normalized updates converge faster as they keep increasing the margin despite small gradients (Nacson et al., 2019). While the setting studied here is separable, we still observe the separation across class frequencies on problems that are not separable, either because the problem has examples with different output for the same inputs, as in Figure 1, or when adding regularization, as in Appendix D.3. The novel element is that the convergence of gradient flow strongly depends on the class frequencies $\pi$, while the convergence of sign descent is independent of the class frequencies.

This setting is admittedly oversimplified and does not capture some of the features observed in our experiments. For example, in Theorem 3, the loss is monotonically decreasing for all classes. This no longer holds once we introduce a bias term and the loss from low-frequency classes will instead first increase, as can be seen for example in Figure 4. This setting is also biased towards sign descent, as the inputs are aligned with the basis vectors. Finally, the problem is inadequate to study large step-sizes, as it can be solved in one large step. On data with non-orthogonal classes, large step-sizes would lead to training instabilities and oscillations in the loss of frequent classes, as can be seen in Figures 2 to 5. Nevertheless, this result formally establishes the benefit of sign-based updates and we believe it captures the key difficulty encountered by GD under heavy-tailed class imbalance.

## 4    Discussion and limitations

**Interaction with stochasticity.** Our experiments include both stochastic and deterministic training regimes and show that stochasticity is not the cause of the slow performance of SGD on low-frequency classes, as it already appears between full batch GD and Adam. This observation is consistent with prior work showing that the performance gap between SGD and Adam on language transformers already appears with deterministic training (Kunstner et al., 2023). However, we do not attempt to quantify the interaction between stochasticity and class imbalance and leave it for future work.

**Training performance vs. generalization.** Our main focus is on optimization performance. Our observations need not generalize to the validation loss, especially in settings prone to overfitting, as good training performance may lead to overfitting on classes with few samples (Sagawa et al., 2020). However, some form of memorization might be needed in long-tailed settings (Feldman, 2020), and if SGD cannot even fit the training data, generalization cannot be good. On the transformer of Figure 1, we observe similar dynamics across frequencies on the validation loss, shown Appendix B.2. Training dynamics on the empirical and population loss are also often similar, particularly early in training (see, e.g., Nakkiran et al., 2021; Ghosh et al., 2022), and the one-pass training regime commonly used in large language models might mitigate those issues by blurring the line between train and test loss.

**Additional difficulties due to text data.** We study the effect of the distribution of the classes, the *next* token to be predicted, but other optimization difficulties might arise from the heavy-tailedness of text data. For example, the sequence of tokens used as inputs to the embedding layer are also heavy-tailed. This imbalance might lead to slow progress for rare tokens with GD, giving another potential cause for a performance gap. With stochastic training, this imbalance leads to sparse updates

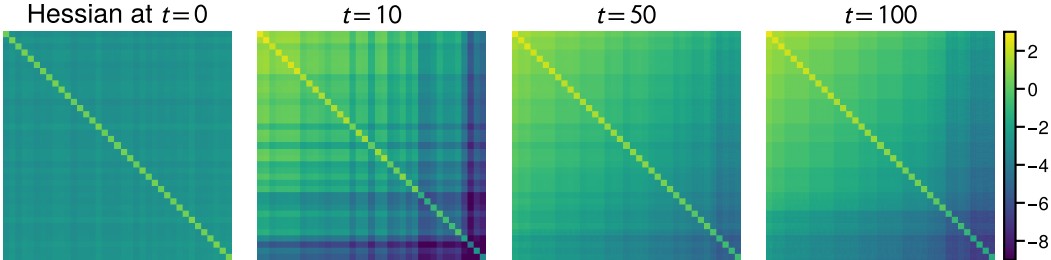

Figure 9: **The diagonal Hessian blocks are orders of magnitude larger than off-diagonal blocks.** Showing the magnitude of a subset of the Hessian blocks ($\log_{10}(|\text{Tr}(\nabla^2_{ij}\mathcal{L})|)$) for a $[160 \times 160]$ subset of the Hessian, sampling 40 classes log-uniformly and 40 input dimensions uniformly.

to the embedding layer, close to the setting that initially motivated AdaGrad (Duchi et al., 2011) and follow-up works attempting to correct this frequency imbalance in the input tokens (Défossez and Bach, 2017; Li et al., 2022). Beyond the inputs, full sentences (Williams et al., 2015) and latent rules or mechanisms required to understand a paragraph (Michaud et al., 2023) may also display heavy tails, and Adam could be beneficial if those are captured by intermediate layers (e.g., Meng et al., 2022; Wang et al., 2023; Bietti et al., 2023). The choice of tokenization has also been shown to impact downstream performance, which has been attributed to the lack of samples on rare tokens (Gowda and May, 2020) and the improved efficiency of more uniform tokenizers (Zouhar et al., 2023). Our results indicate that tokenization also has a large impact on optimization performance.

**Difficulties due to architectures.** Beyond the class distribution, additional optimization difficulties may arise from the architectures, due to depth, signal propagation (Noci et al., 2022; He et al., 2023), vanishing gradients and higher order derivatives (Liu et al., 2020; Orvieto et al., 2022). The simplified transformer of Ahn et al. (2023) also exhibits many of the difficulties observed in the literature on regression instead of a classification problem. However, a phenomenon similar to the assignment mechanism could still explain the benefit of Adam. The oscillations in the loss observed at the feature level by Rosenfeld and Risteski (2023) suggests a link between subsets of the samples and subsets of the parameters. For example, if a convolution filter detects a specific background color and captures a specific feature of the data, the magnitude of the gradient and Hessian at intermediate layers could be influenced by the relative frequency of the feature in the data, leading to another form of imbalance.

**Recent ablations on the benefit of Adam for language transformer.** Parallel to our work, recent investigations have looked into the benefits of Adam on language transformers. Zhang et al. (2024a) argue that the Hessian has a block-diagonal structure, with similar magnitude within blocks but very different magnitudes across blocks, and that Adam may improve performance by using a different step-size for different blocks. This hypothesis is supported by recent ablations studies. Zhang et al. (2024b) show that the element-wise preconditioning in Adam is not necessary and can be replaced by a single parameter across such blocks while maintaining performance, which they coin Adam-mini. Similarly, Zhao et al. (2024) show that the performance of Adam can be recovered by training most of the network with (S)GD, except for the last layer and LayerNorm parameters. Both approaches still need to treat the last layer separately, either using a step-size for each row $\mathbf{w}_c$ of the last layer in the case of Zhang et al. (2024b) or by using Adam to train the last layer in the case of Zhao et al. (2024). These observations complement our approach, which focuses on the impact of heavy-tailed class imbalance on the last layer, and are consistent with our conclusion that one of the main benefit of Adam is to counteract the slow progress on rare classes by preconditioning the last layer.

## 5 Conclusion

We have shown that heavy-tailed class imbalance leads to a performance gap between (S)GD and Adam. This effect is reproducible across architectures and data types, but is most salient on language tasks which naturally exhibit heavy-tailed imbalance. As vision tasks are typically more uniform, imbalance is a key differentiating feature of the training difficulties in language tasks. The correlation between entries of the gradient and Hessian that occurs due to class imbalance provides justification for the intuition that Adam-like algorithms "adapt to curvature". We provide an explanation for how this correlation arises during training through the assignment mechanism and prove on a simplified setting that gradient descent performs poorly on low-frequency classes while sign descent does not.

## Acknowledgements

We thank Greg d'Eon, Aaron Mishkin, Victor Sanches Portella, and Danica Sutherland for useful discussions and comments on the manuscript. This research was supported by the Canada CIFAR AI Chair Program, the Natural Sciences and Engineering Research Council of Canada (NSERC) through the Discovery Grants RGPIN-2022-03669, and was enabled by the support provided by the BC DRI Group and the Digital Research Alliance of Canada (`alliancecan.ca`).

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

# Supplementary Material

## Table of Contents

## A  Experimental details

This section documents the datasets, models, software, and experimental setup. The code is available at https://github.com/fkunstner/class-imbalance-sgd-adam.

### A.1  Datasets

- **WikiText-103** (Merity et al., 2017), using sequences of $1\,024$ tokens and the BPE tokenizer (Sennrich et al., 2016), with a vocabulary of size $50\,608$.

- **WikiText-2** (Merity et al., 2017) is used in Appendix B.1 to illustrate that other combinations of datasets and tokenizers lead to heavy-tailed distributions.

- **PTB** (Marcus et al., 1993), using sequences of 35 tokens built from a word-based tokenizer (`basic_english` provided by `torchtext`), for a vocabulary of size $9\,920$. For deterministic runs, we use the validation set as a reduced training set, labeled **TinyPTB**.

- **MNIST** (LeCun et al., 1998).

- **ImageNet** (Deng et al., 2009).

### A.2  Custom datasets

- **The Random Heavy-Tailed Labels dataset** is a synthetic dataset exhibiting heavy-tailed class imbalance. The number of samples per class and the number of classes are picked to approximate a power-law distribution. We create $m$ "groups" of classes, where each class within a group has the same relative frequency;

$$\underbrace{1 \text{ class with } 2^m \text{ samples},}_{\text{group 1}} \quad \underbrace{2 \text{ classes with } 2^{m-1} \text{ samples},}_{\text{group 2}} \quad \dots, \quad \underbrace{2^{m-1} \text{ classes with } 2 \text{ samples}.}_{\text{group } m}$$

  The inputs are drawn from a uniform distribution on $[0, 1]$, independently of the class label. The inputs are in $d = (m + 1)\, 2^m$ dimensions, the number of samples is $n = m\, 2^m$ and the number of classes is $c = 2^{m+1} - 1$. We use two variants of the datasets; a large one in Figure 4, Appendix E ($m = 11, n = 22\,528, d = 24\,576, c = 4\,095$) and a small one in Appendix D ($m = 8, n = 2\,048, d = 2\,304, c = 511$).

- **The Barcoded MNIST dataset** is a modified variant of MNIST. We start with 50k examples from the original MNIST dataset across 10 classes, and create $51\,150$ ($5 \times (10 \times 2^{10} - 1)$) new

images. The new examples are copies of existing image with an added "barcode", a 10-bit number encoded in a corner of the image, as in the examples below. The class label is a combination of the original class and this barcode.

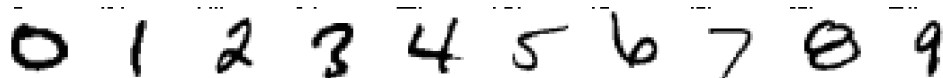

The **Barcoded-only** dataset contains $10 \times 2^{10}$ classes with 5 samples each. To obtain an imbalanced dataset, we combine the barcoded images with the original samples from the MNIST dataset to get $101\,200$ examples spread across $10\,250$ ($10 \times 2^{10} + 10$) classes classes; $10\,240$ with 5 examples per class and 10 classes with $\approx 5k$ examples per class, labeled **MNIST+Barcoded**

- **The Heavy Tailed ImageNet** dataset is a subset of ImageNet (Deng et al., 2009), subsampled to exhibit heavy-tailed class imbalance. We sort the original 1000 classes by frequency and sample $\lceil 1300/k \rceil$ images from the $k$th class, leading to $n = 10\,217$ samples.

- **The Small ImageNet** dataset is a uniform subset of ImageNet to contrast the with the heavy tailed variant. We sample 10 images per class to get $n = 10\,000$ samples.

### A.3   Models

- **The 2-layer transformer** used in Appendix B.3 is a transformer Vaswani et al. (2017), based on the PyTorch implementation of `TransformerEncoderLayer` (Paszke et al., 2019).

$$\text{Embedding} \rightarrow 2\times [\text{Attention} \rightarrow \text{Linear} \rightarrow \text{ReLU} \rightarrow \text{Linear}] \rightarrow \text{Classifier}.$$

  The model includes LayerNorm, dropout, and skip connections (He et al., 2016; Ba et al., 2016; Srivastava et al., 2014). The embedding dimension and width of the linear layers is 1000 and the attention modules use 4 heads.

- **The simplified transformer** used in Figure 5 and Appendix B.3 does not use encoder blocks, and only uses attention:

$$\text{Embedding} \rightarrow \text{Attention} \rightarrow \text{Classifier}.$$

  We remove LayerNorm, dropout, and the block [Linear $\rightarrow$ ReLU $\rightarrow$ Linear] containing the non-linearity. In Figure 5, we freeze the embedding and attention layers at initialization, and only the last classification layer is trained. The model is then a linear model on a fixed feature transformation.

- **The GPT2-Small** model (Radford et al., 2019) is used in Figure 1. The blocks includes LayerNorm, residual connections, and dropout on the embedding and dense layers. We use sinusoid positional encodings as in the transformer architecture (Vaswani et al., 2017). The embedding dimension is 768, the width of the intermediate layers is 3072, and we use 12 encoder blocks with 12 self attention heads.

- **The convolutional network** used in Figure 2 and Appendix C is a 2-layer convolution

$$\text{Conv} \rightarrow \text{Relu} \rightarrow \text{MaxPool} \rightarrow \text{Conv} \rightarrow \text{Relu} \rightarrow \text{MaxPool} \rightarrow \text{Linear}$$

- **The linear model** used in Figures 4 and 7 and Appendix E uses a bias vector.

- **The ResNet18** model (He et al., 2016) is used in Figure 3. Additionally, a variant replacing the BatchNorm layers with LayerNorm is used in Appendix C.

- **The SimpleViT** model (Beyer et al., 2022) used in Appendix C follows the architecture of a ViT-S/16 (Touvron et al., 2021), based on the `vit-pytorch` implementation (https://github.com/lucidrains/vit-pytorch v1.6.5).

### A.4   Training procedures

Our primary focus is on the performance of the optimizers on the training error, using the simplest training procedure possible. We use a constant step-size throughout training, set by grid search. We start with a sparse grid of powers of 10 $[10^{-6}, 10^{-2}, ..., 10^1]$ and increase the density to half-powers around the best step-size. The step-size is selected to minimize the maximum over 3 seeds of the training loss at the end of training. For some settings, this selection still produces runs that are unstable; the training loss is the smallest at the end but oscillates a lot during training, reaching

loss values that are orders of magnitude worse than at initialization. For those runs, we use the next smaller step-size, which has similar performance at the end but is more stable. We use the following batch sizes with gradient accumulation (computing the gradient through multiple passes)

- The large transformer experiment in Figure 1 uses mini-batches of 512 sequences of 1024 tokens.
- The stochastic experiments with a smaller transformer in Appendix B.3 uses mini-batches of 512 sequences of 35 tokens.
- Both ResNet18 variants and the Simple Vision Transformer were trained using mini-batches of 1024. The training images were normalized and randomly cropped to $224 \times 224$ pixels as is standard for ImageNet training.
- Other experiments use the entire dataset to compute updates

Our experiments ran on a cluster using a mix of A100, P100, V100, and H100 GPUs. The large scale experiment in Figure 1 took 3 days on a H100, while all other experiments ran in 2–8 hours. The total amount of compute used for this project is $\approx 3$ GPU-years, including preliminary experiments.

## A.5 Optimization algorithms

Given momentum buffers $m_t$ initialized at $m_0 = 0$ and a (possibly) stochastic gradient $\tilde{g}_t$, we implement the update of GD, normalized GD and sign descent with heavy-ball momentum as

$$
\begin{aligned}
m_t &= \beta m_{t-1} + d_t, \\
x_{t+1} &= x_t - \alpha m_t,
\end{aligned}
\qquad \text{with } d_t = \begin{cases} \tilde{g}_t & \text{for gradient descent,} \\ \tilde{g}_t / \|\tilde{g}_t\|_2 & \text{for normalized GD,} \\ \operatorname{sign}(\tilde{g}_t) & \text{for sign descent.} \end{cases}
$$

For GD and Adam, we use the standard implementation in PyTorch (Paszke et al., 2019). For all algorithms, we use either momentum with $\beta = 0.9$ ($\beta_1 = 0.9$ for Adam) or no momentum ($\beta = 0$, $\beta_1 = 0$), indicated by solid lines and the legend (+m) for runs with momentum, and dashed lines and the legend (-m) for runs without momentum.

## A.6 Summary of settings used

Table 1: Summary of models, datasets and batch-size used

| Model | Dataset | Batch size | Used in |
| --- | --- | --- | --- |
| GPT2-Small | WT103 | 512 | Figure 1 and Figure 11 |
| 2-layer transformer | PTB | 512 | Figures 12, 25 and 31 |
| 1-layer transformer | TinyPTB | Full | Figures 13 and 23 |
| 1-layer transformer | TinyPTB | Full | Figure 5 (last layer only) |
| CNN | Barcoded MNIST | Full | Figure 18 |
| CNN | MNIST | Full | Figures 2 and 18 |
| CNN | MNIST+Barcoded | Full | Figures 2, 18, 24, 25 and 29 |
| Linear | Random HT labels, m=11 | Full | Figures 4, 7, 22, 25, 26, 32 and 33 |
| Linear | Random HT labels, m=7 | Full | Figures 19 and 20 |
| Simple ViT | ImageNet | 1024 | Figure 16 |
| ResNet18 | Small and HT ImageNet | 1024 | Figures 3, 25 and 30 |
| ResNet18+LN | Small and HT ImageNet | 1024 | Figure 15 |
| Simple ViT | Small and HT ImageNet | 1024 | Figure 17 |

# B  Language problems

This section provides additional ablations on language models, showing that the impact of class imbalance holds across models of different sizes and using deterministic updates.

B.1 shows that the heavy-tailed distribution in text data occurs across datasets and tokenizers.

B.2 shows that the imbalanced training speed across frequencies translates to the validation loss.

B.3 shows that the imbalance training speed across frequencies and the gap between SGD and Adam can be reproduced with smaller transformers. This effect also appears when training only the last layer, and in the deterministic setting, comparing GD and Adam.

## B.1  Class distribution for common datasets and tokenizers

Figure 10 provides additional examples of the heavy-tailed distribution of tokens using the basic english tokenizer in `torchtext` (Paszke et al., 2019), Byte-Pair Encoding (BPE, Sennrich et al., 2016; Gage, 1994) and Unigram (Kudo, 2018) on the PTB and WikiText-2 datasets. The relationship between the relative frequency rank $k$ and and the relative frequency $\pi_k$ is roughly $\pi_k \propto 1/k$.

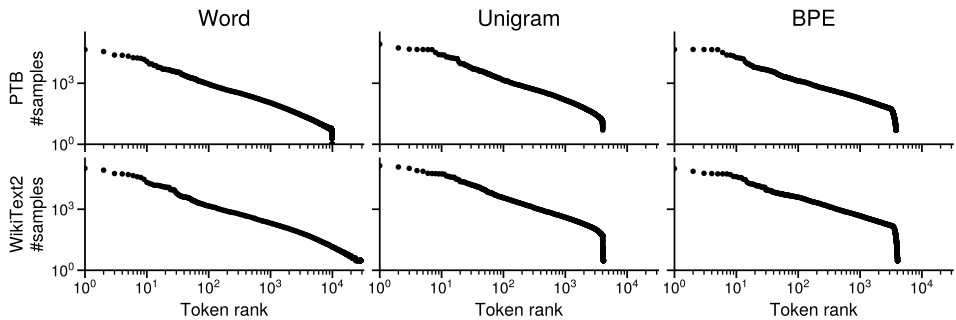

Figure 10: **Different tokenizers and datasets lead to heavy-tailed token distributions.** Comparison of word and subword tokenization (BPE, Unigram) on the PTB and WikiText2 datasets.

## B.2  Effect of class imbalance on validation loss

In Figure 11, we show the validation error on the same problem as Figure 1, training GPT2-Small on WikiText-103. The validation loss exhibits the same separation across class frequencies, and the faster progress of Adam on low-frequency classes is also visible. While this trend does not hold for all the settings we investigate, as some settings use smaller datasets and deterministic training to isolate the source of the training difficulties, the benefit of Adam on low-frequency classes does not immediately lead to overfitting.

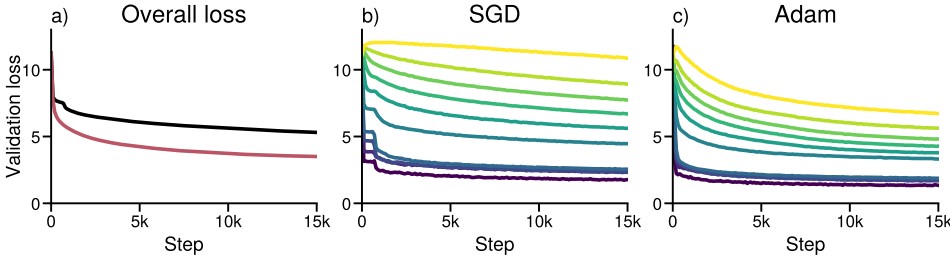

Figure 11: **The class-separation behavior of Figure 1 holds on the validation loss.** Same experiment as Figure 1, training GPT2-Small on WikiText-103, but showing the validation loss. (a) Distribution of the classes sorted by class frequency, split into groups corresponding to ≈10% of the data. (b) Overall validation loss. (c, d) Validation loss for each group using SGD and Adam. SGD makes little to no progress on low-frequency classes while Adam makes progress on all groups. (b) is the average of (c, d) for the respective optimizer.

## B.3 Smaller transformers and deterministic training

In Section 2.3, we argued that the qualitatively different behavior on low-frequency classes between SGD and Adam in Figure 1 is not due to stochasticity. In this section, we provide additional results showing that this behavior appears across multiple batch sizes on language transformers of different sizes and that it can be reproduced in the deterministic setting.

In Figure 12, we show that a similar qualitative behavior appears when training a smaller model (2-layer transformer) on a smaller dataset (PTB). In Figure 13, we repeat the experiment with a 1-layer transformer, trained in full batch on TinyPTB (the validation set of PTB). The separation between GD and Adam on low-frequency classes in the deterministic settings is also visible in Figures 2, 4, 5 and 7 in the main paper. These results indicate that stochasticity it is not necessary to reproduce the behavior observed in Figure 1. Finally, we repeat the experiment but freeze all the layers except the last, and still observe this behavior in Figure 14.

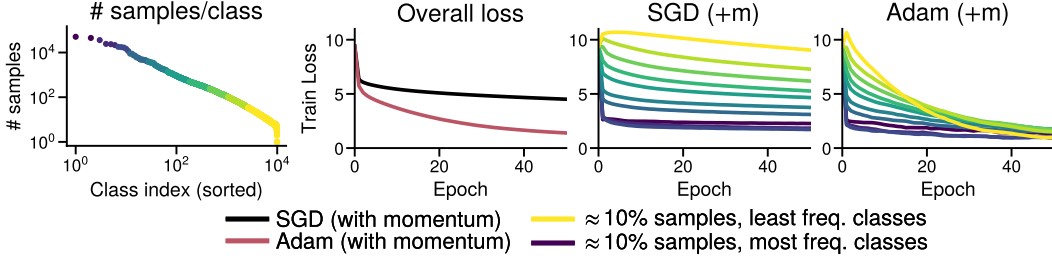

Figure 12: **Similar behavior as Figure 1 on a smaller problem.** Training a 2-layer transformer on PTB with Adam and SGD using larger batch-sizes. As in Figure 1, SGD makes little to no progress on low-frequency classes while Adam makes progress on all subsets. Subplots: **(1)** Distribution of the classes and subsets of the data sorted by class frequency, each corresponding to ≈10% of the samples. **(2)** Overall training loss. **(3, 4)** Training loss for each subset for SGD and Adam.

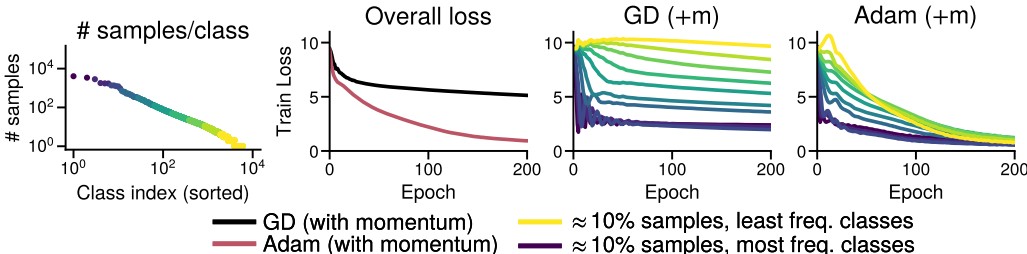

Figure 13: **Similar behavior as Figure 1 on a one-layer transformer with deterministic updates.** Trained on TinyPTB. Subplots: **(1)** Distribution of the classes and subsets of the data sorted by class frequency. **(2)** Overall training loss. **(3, 4)** Training loss for each subset for GD and Adam.

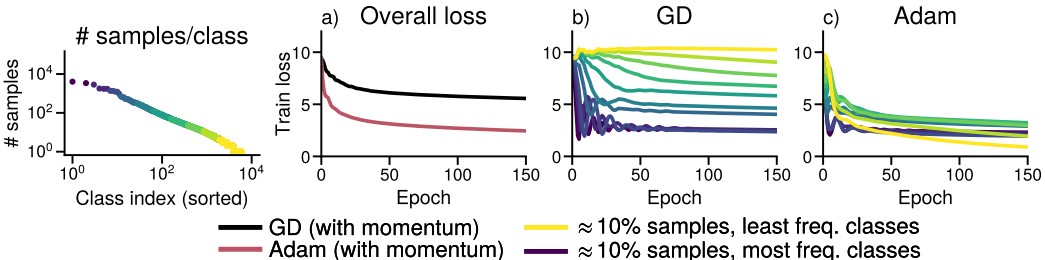

Figure 14: **Similar behavior as Figure 1 when training only the last layer.** Training the last layer of a 1-layer transformer on PTB with Adam and GD with deterministic updates. Subplots: **(1)** Distribution of the classes and subsets of the data sorted by class frequency. **(2)** Overall training loss. **(3, 4)** Training loss for each subset for GD and Adam.

# C Vision problems

This section gives additional results on vision tasks to complement Section 2.1.

- Figure 15 shows a similar behavior on a ResNet18 with LayerNorm instead of BatchNorm.
- Figure 16 shows a similar behavior with a vision transformer.
- Figure 18 confirms that GD can solve the Barcoded MNIST variant without imbalance.

## C.1 ResNet18 with LayerNorm

In Figure 15, we use the same settings Figure 3. training a ResNet18 on a uniform and unbalanced subset of ImageNet, but replace the normalization layers with LayerNorm (Ba et al., 2016) instead of BatchNorm (Ioffe and Szegedy, 2015). We observe a similar pattern as in Figure 3. Although Adam slightly outperforms SGD on the uniform dataset, the performance gap grows on the imbalanced one.

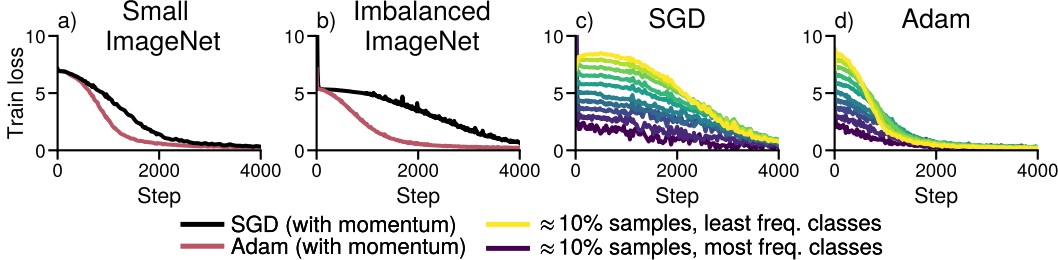

Figure 15: **Adam outperforms SGD on ResNet with LayerNorm under heavy-tailed imbalance.** (a) Performance on a uniform subset of ImageNet (b) and on an imbalanced subset with class frequencies $\pi_k \propto 1/k$. (c, d) Performance of GD and Adam across frequencies.

## C.2 Vision Transformers

In Figure 16, we train a vision transformer on the ImageNet dataset, without subsampling, to confirm that the training behavior is similar. While vision transformers might require more data or regularization than their ResNet counterparts to achieve comparable generalization performance, the optimization problem does not appear to be more difficult for SGD than for Adam.

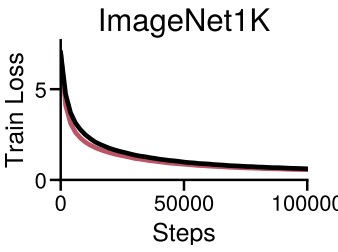

Figure 16: **Adam and SGD perform similarly training a Vision Transformer with balanced Classes.** Training loss on the full ImageNet dataset (without subsampling). There is little performance in training performance.

In Figure 17, we train the same vision transformer on the uniform and imbalanced subsets of ImageNet. As in prior experiments with vision data, the performance of Adam appears unaffected by the change in class frequencies while the performance of SGD degrades.

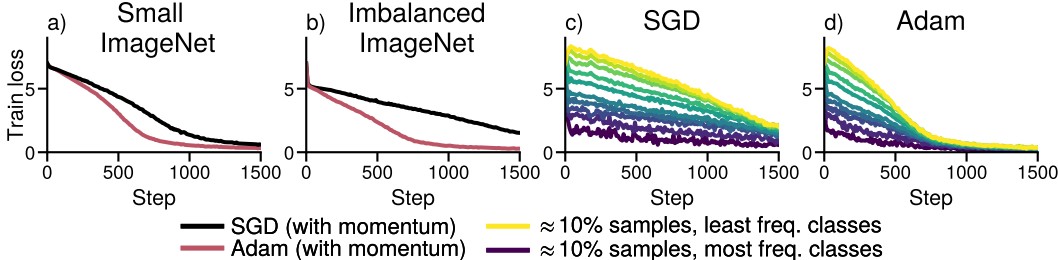

Figure 17: **Adam outperforms SGD on vision transformer under heavy-tailed imbalance.** (a) Performance on a uniform subset of ImageNet (b) and on an imbalanced subset with class frequencies $\pi_k \propto 1/k$. (c, d) Performance of GD and Adam across frequencies.

### C.3 Sanity check on Barcoded MNIST

Figure 2 in Section 2.1 showed that the performance gap between GD and Adam on the imbalanced variant of MNIST with barcoded images is larger than on plain MNIST. In this section, we verify that the training difficulties encountered on the CNN on the imbalanced MNIST dataset of Figure 2 are indeed due to class imbalance. As we create new images and new classes by adding a barcode in the corner of existing images, it could be that the dataset becomes harder to fit.

In Figure 18, we run Adam and GD to train the same network on the MNIST dataset only, the barcoded-only subset of the imbalanced MNIST and the combination of the two, leading to an imbalanced dataset. While Adam is faster GD on the barcoded-only dataset, both algorithms reach negligible error within 200 steps. In contrast, on the combined imbalanced dataset MNIST+Barcoded, GD fails to make progress on the low-frequency classes and stalls.

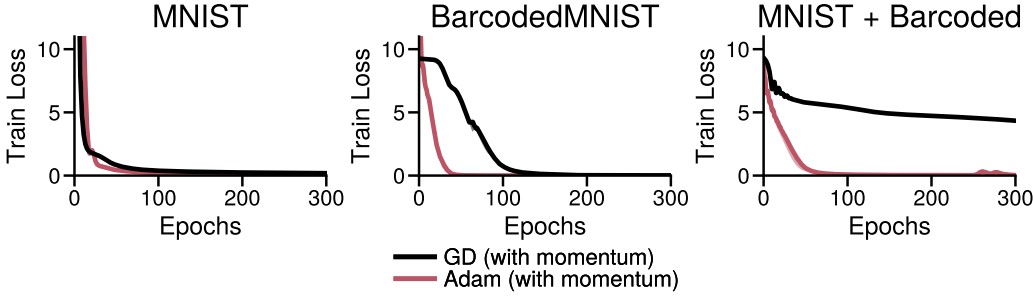

Figure 18: **GD optimizes on balanced barcoded data.** Training a CNN on only the barcoded portion of the data, which has balanced classes. While Adam is slightly faster, both optimizers reach negligible error within 200 steps. As the level of imbalance is increased, GD performs increasingly worse than Adam.

# D Linear models

Section 2.2 showed that GD is already slow on linear models. We give additional details here.

D.1 discusses the impact of the distribution of the inputs, as it is possible to construct problems exhibiting class imbalance without negatively impacting training.

D.2 shows that while (S)GD appears stuck in some experiments, it is not due to being stuck in a local minima. It eventually converges, although very slowly, if run long enough.

D.3 shows that while some of our datasets are separable, leading to weights going to $\infty$, class imbalance also impacts optimization when the weights remain small, e.g. when using l2 regularization.

## D.1 Impact of input distribution

Imbalance alone is not sufficient to induce slow performance of GD on low-frequency classes. It is possible to generate a dataset with heavy-tailed class imbalance where GD fits all classes fast, by making all inputs $\mathbf{x}_i$ (close to) orthogonal, $\langle \mathbf{x}_i, \mathbf{x}_j \rangle \approx 0$ for $i \neq j$. If all samples are orthogonal, $\langle \mathbf{x}_i, \mathbf{x}_j \rangle = 0 \ \forall i \neq j$, a decomposition similar to that used in the proof of Theorem 3 shows that each sample is learned independently of the other, and class frequency has no impact. However, completely orthogonal data is rare. In the last layer of neural networks, we expect samples from the same class to be mapped to similar representation (Papyan et al., 2020), a phenomenon also observed under class imbalance (Thrampoulidis et al., 2022). Using a bias term also increases alignment between samples, as it is equivalent to adding a dimension where each sample has the same value.

In the setting of Theorem 3, class imbalance has an impact because samples from the same class are collinear, even though samples from separate classes are orthogonal. A more realistic mixture model where samples from the same class are aligned ($|\langle \mathbf{x}_i, \mathbf{x}_j \rangle| > \delta$ if $y_i = y_j$) but independent otherwise ($|\langle \mathbf{x}_i, \mathbf{x}_j \rangle| \leq \epsilon$ if $y_i \neq y_j$), as the setting of Feldman (2020) would also exhibit class separation. The class imbalance appears in Figure 4 because we draw the inputs from a high-dimensional uniform distribution on $[0,1]^d$, ensuring that for any two samples $\mathbf{x}_i, \mathbf{x}_j$, $\langle \mathbf{x}_i, \mathbf{x}_j \rangle > 0$. If the data was sampled from $\mathcal{N}(0,1)^d$ in sufficiently high dimension, the samples would be independent enough to avoid the slowdown due to class imbalance. We illustrate this in Figure 19, where we use a smaller synthetic data with inputs drawn from $\mathcal{N}(1,1)$ and $\mathcal{N}(0,1)$. The zero-mean data is be approximately orthogonal as $d > n$ and does not exhibit a slow progress on low-frequency classes.

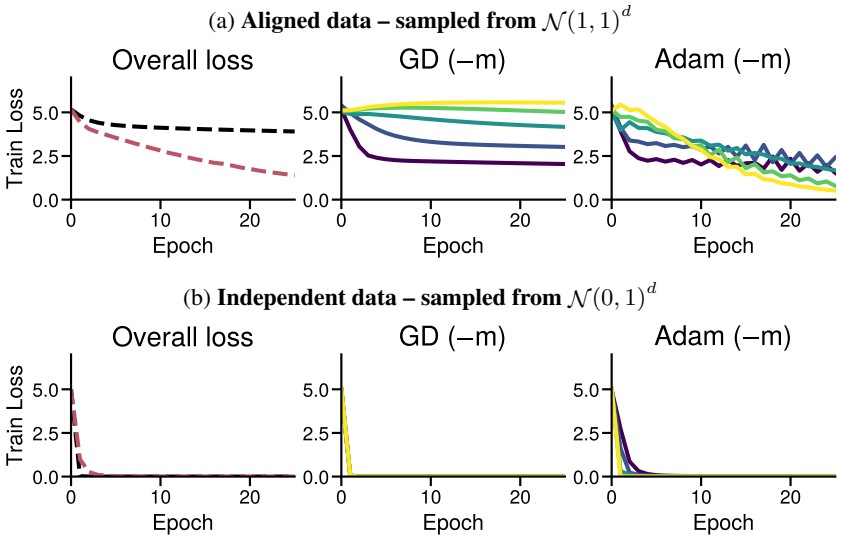

Figure 19: **The distribution of the inputs can have a large impact on optimization.** Linear model on the Random Heavy-Tailed Labels dataset, with Inputs sampled from $\mathcal{N}(1,1)$ (a) and $\mathcal{N}(0,1)$ (b).

The behavior of GD on aligned data appears to be a better representation of the behavior of GD on language transformers, as we observe a performance separation per class frequency on GD, even when tuning only the last layer of a language transformer in Figure 5. Although the embedding weights are initialized to be zero-mean Gaussian noise, the representation of the tokens in a transformer are aligned, and this alignment increases with depth (Noci et al., 2022, e.g.).

## D.2 An early iteration problem

As GD is slower than Adam at fitting the low-frequency classes, it might seem that GD does not fit the low-frequency classes at all. But when run for longer, GD converges and fits all classes. We show this behavior on the linear model and the CNN on imbalanced MNIST in Figure 20. This highlight that the difference between the algorithms is primarily a difference at the start of training. However, this "start" can be quite long. In the transformer of Figure 1, the average loss on $10\%$ of the data corresponding to the least frequent classes is still higher than at initialization after 15k steps.

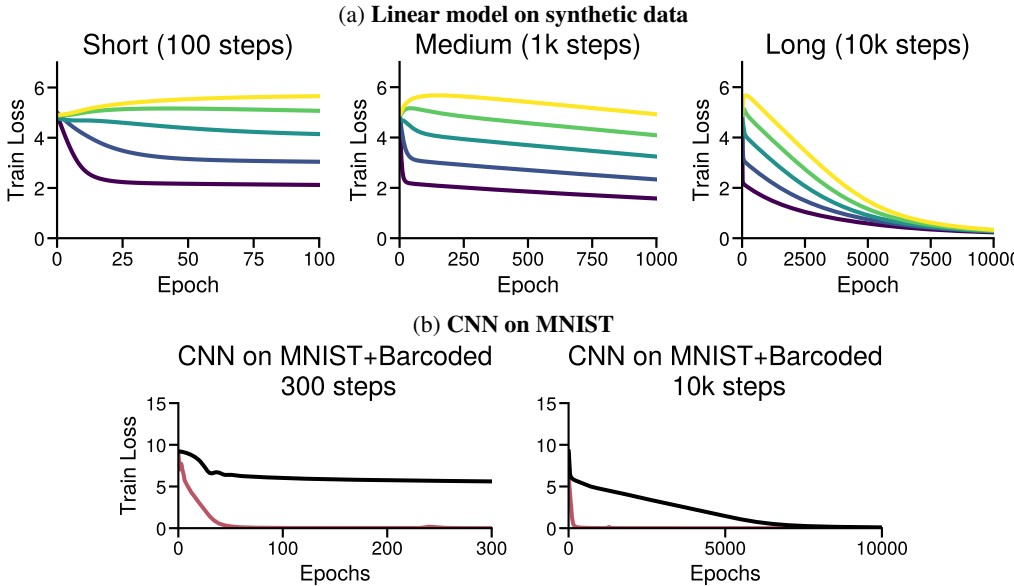

Figure 20: **Training with GD eventually drives the loss down for all classes.** Using the same step-size for different horizons (100, 1k, 10k). GD eventually drives the loss down for all classes, but the loss for the least-frequent classes only decreases below its value at initialization after 1k steps. (a) Linear model on synthetic data, (b) CNN on MNIST.

### D.3 Impact of regularization

The data used with the linear model of Figure 4 is separable, meaning the predicted probabilities for the correct class will converge to 1 while the magnitude of weights go to $\infty$. This might lead to concerns that the observed behavior is tied to the weights growing without bounds. In Figure 21, we show that the gap between GD and Adam still appears with regularization limiting the magnitude of the weights. However, as regularization is increased, the L2 penalty makes it difficult to fit low-frequency classes, the problem looks more like $\lambda\frac{1}{2}\|\cdot\|^2$, and the gap between the methods disappears.

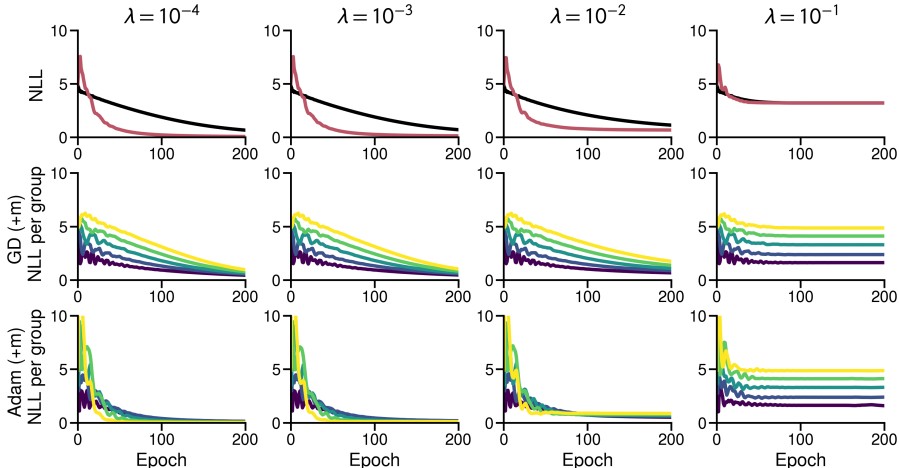

Figure 21: **The separation between GD and Adam still appears when using $L_2$ regularization.** Using varying levels of regularization $\lambda$ on the linear model of Figure 4. The plots show the negative log-likelihood and do not include the $L_2$ penalty.

# E  Alternative optimizers

Figure 5 in Section 2.3 we compared GD and Adam to normalized GD and sign descent on the last layer of a one-module transformer on TinyPTB, showing that Adam and sign descent perform similarly. We repeat this experiment on other settings here to confirm that sign descent leads to similar benefits as Adam on low-frequency classes, and that changing the direction, as in sign descent, has more impact than just changing the magnitude, as in normalized GD.

We also observe this behavior on the following problems:

- Figure 22: A linear model on **Random Heavy-Tailed Labels**, as in Figure 4.
- Figure 23: A one-module transformer on **TinyPTB**, as in Figure 13, training all layers.
- Figure 24: A CNN on **MNIST+Barcoded**, as in Figure 2.

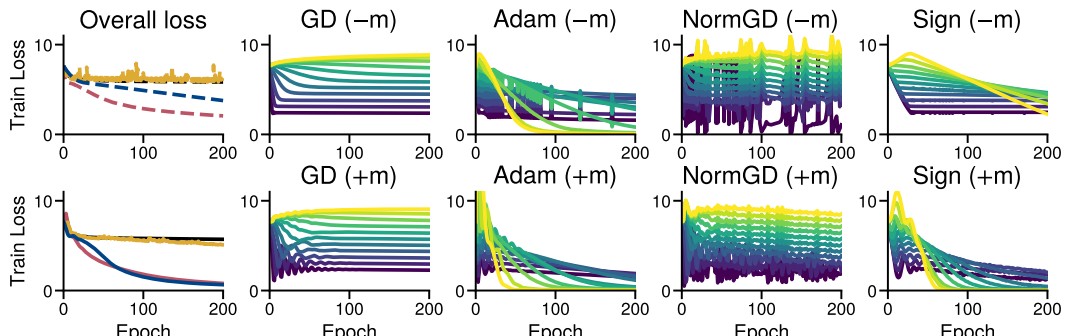

Figure 22: **All optimizers on the linear model of Figure 4.**

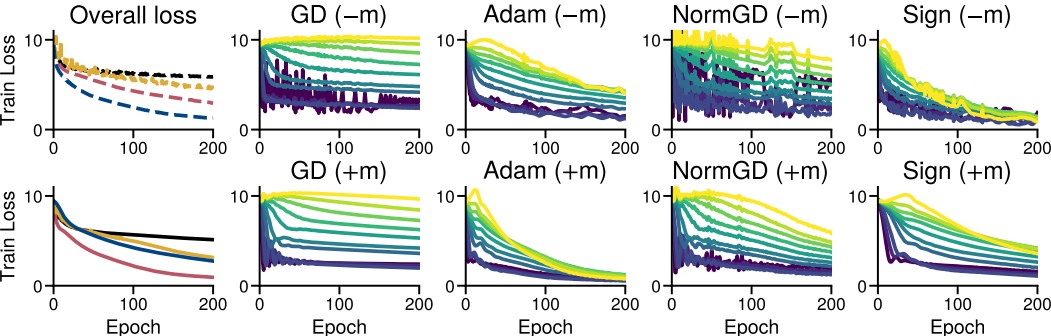

Figure 23: **All optimizers on the transformer of Figure 13.**

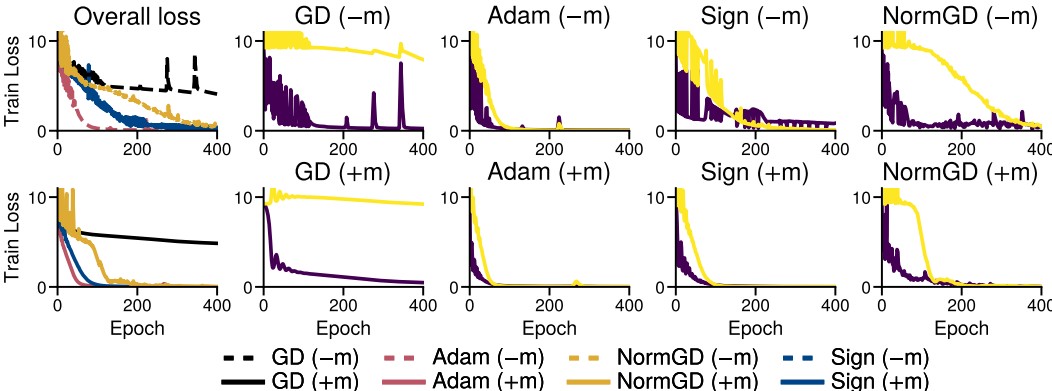

Figure 24: **All optimizers on the CNN of Figure 2.** First column: Overall training loss. Remaining: Loss by frequency groups for each optimizer, with and without momentum (+m, bottom/−m, top).

### E.1 Up-weighting low-frequency classes can improve the performance of SGD

To support Section 2.3, we show show that upweighting low-frequency classes helps reduce the performance gap between SGD and Adam on problems with heavy-tailed class imbalance, providing evidence that the optimization difficulties are associated with class imbalance.

While reweighting the loss of samples from class $k$ by $1/\pi_k$ to address the class imbalance seems intuitive, optimizing the reweighted loss is no longer guaranteed to lead to progress on the original loss, especially if the weights are large. Indeed, we find that on some problems this reweighting does not improve performance (although SGD and Adam perform similarly on the reweighted loss, not shown). However, the less extreme reweighting of $1/\sqrt{\pi_k}$ appears to consistently outperform SGD.

In Figure 25, we run SGD on the reweighted loss with the two weighting schemes, $1/\pi_k$ and $1/\sqrt{\pi_k}$ and plot its performance on the original, unweighted loss. We compare the performance of the two reweighting schemes with SGD and Adam, all with momentum, on the following 4 problems.

- The small transformer on PTB in Figure 12 (stochastic training)
- The Linear model on synthetic data in Figure 4 (deterministic training)
- The CNN on MNIST+Barcoded dataset in Figure 2 (deterministic training)
- The ResNet18 on the Heavy-Tailed ImageNet dataset in Figure 3 (stochastic training)

We found that the combination of both Adam and reweighting did not improve over running Adam on the original loss and do no include it in Figure 25.

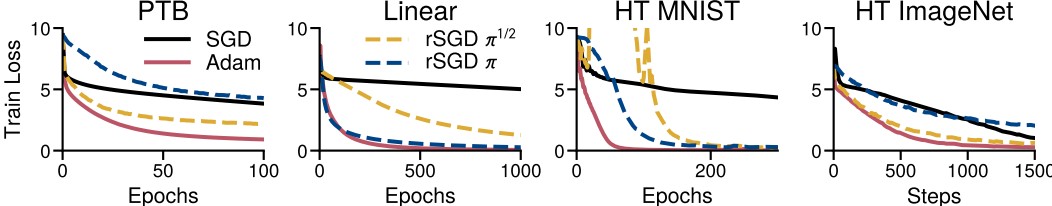

Figure 25: **Reweighting the loss improves the performance of (S)GD on low-frequency classes.** The plots show the unweighted loss, while (S)GD and Adam optimize a reweighted loss. Reweighted (S)GD (r(S)GD) with weights $1/\sqrt{\pi_k}$ consistently outperforms plain SGD, although it can lead to spikes, as on the CNN on the MNIST dataset. Reweighting with weights $1/\pi_k$ is sometimes better (Linear, MNIST) but can be worse (PTB, ImageNet) as it optimizes a different objective. We use deterministic updates for the first 3 problems, labeled Epoch, and stochastic updates for the ResNet18 on heavy-tailed ImageNet.

# F    Dynamics of the gradient and Hessian

This section provides additional details on the dynamics of (S)GD and Adam discussed in Section 3.2.

- Figure 26 shows the dynamics of GD and Adam on the linear model on synthetic data in Figure 4 (deterministic training). This figure complements Figure 7 which shows the dynamics over the path taken by Adam.
- Figure 27 and additionally shows the average predicted probabilities $p$ for each frequency group, showing that the deviation from the linear relationship for rare classes coincides with the predicted probabilities $p$ for those classes going to 1.
- The following figures show the correlation on additional problems, on
    - Figure 28 The GPT2-Small model on WikiText-103 in Figure 1 (stochastic training). This figure complements Figure 8 which shows the dynamics over the path taken by Adam.
    - Figure 29 The CNN on the MNIST+Barcoded dataset in Figure 2 (deterministic training)
    - Figure 31 The small transformer on PTB in Figure 12 (stochastic training)
    - Figure 30 The ResNet18 on the Heavy-Tailed ImageNet dataset in Figure 3 (stochastic training)
- Figure 32 illustrates that this correlation does not hold globally and only emerges throughout training by showing that a *negative* correlation can instead be found by looking at the oppositve path of the path taken by Adam, $-\mathbf{W}_t$ (when $\mathbf{W}_t$ are the iterates generated by Adam).

## F.1    Linear model on synthetic data

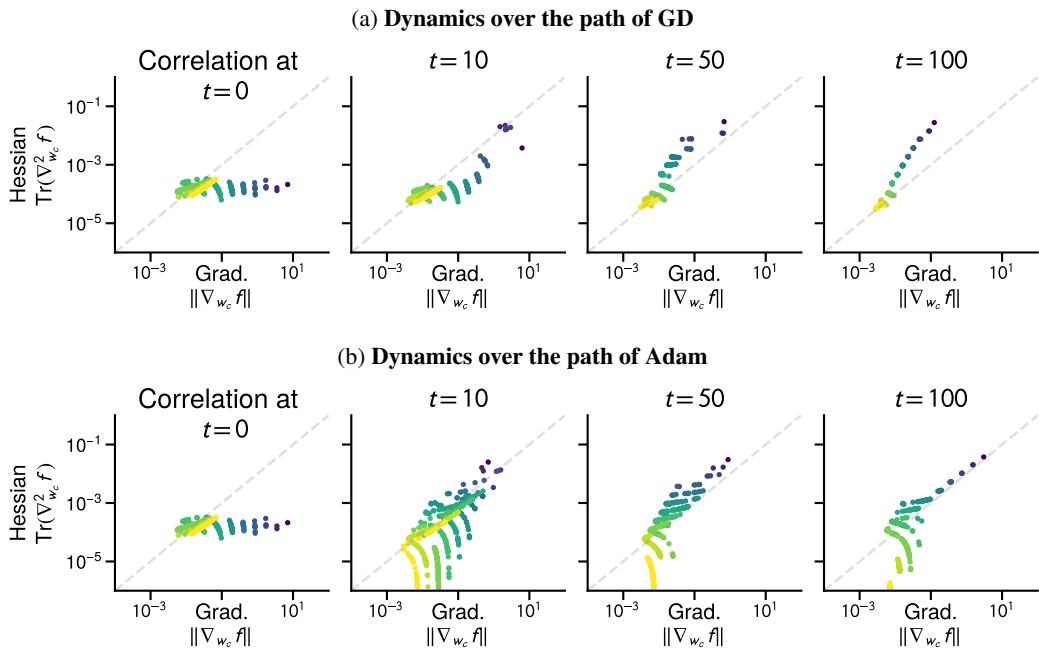

Figure 26: **Evolution of the gradient norm and Hessian trace through optimization.** Taken over the path of GD (a) and Adam (b) on the linear problem of Figure 4. The blocks correspond to the rows $\mathbf{w}_1, ..., \mathbf{w}_c$ of the parameter matrix $\mathbf{W}$. The color indicates the class frequency, showing that lower (higher) frequency classes have smaller (larger) gradient norm and Hessian trace. Figure 26b is a replication of Figure 7, given here for convenience. The deviation from the correlation is explainable by the fact that difference classes are learned at difference speed, leading to a different value of $p$ in Proposition 2, shown in Figure 27. For GD, frequent classes are learned faster than infrequent ones, while for Adam, $p$ is similar among the most frequent groups of classes while $p \to 1$ for the least frequent classes.

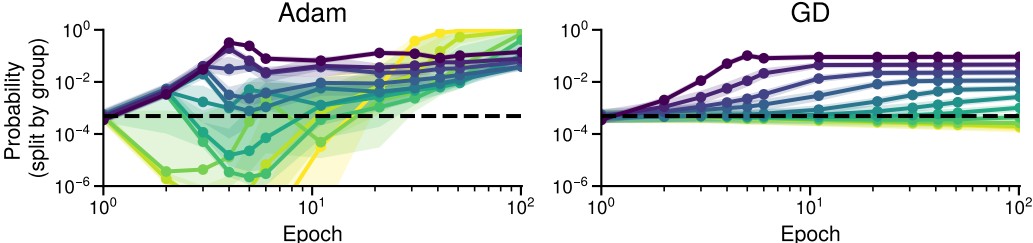

Figure 27: **Evolution of the predicted probabilities for the correct class.** Complement to Figure 26, taken over the path of GD and Adam on the linear problem of Figure 4. For GD, frequent classes are learned faster than infrequent ones, Adam has a similar behavior on the most frequent groups of classes but also increases the predicted probability for the correct class on infrequent groups. The color indicates the class frequency, showing that lower (higher) frequency classes have smaller (larger) gradient norm and Hessian trace.

### F.2  GPT2-Small on WikiText-103

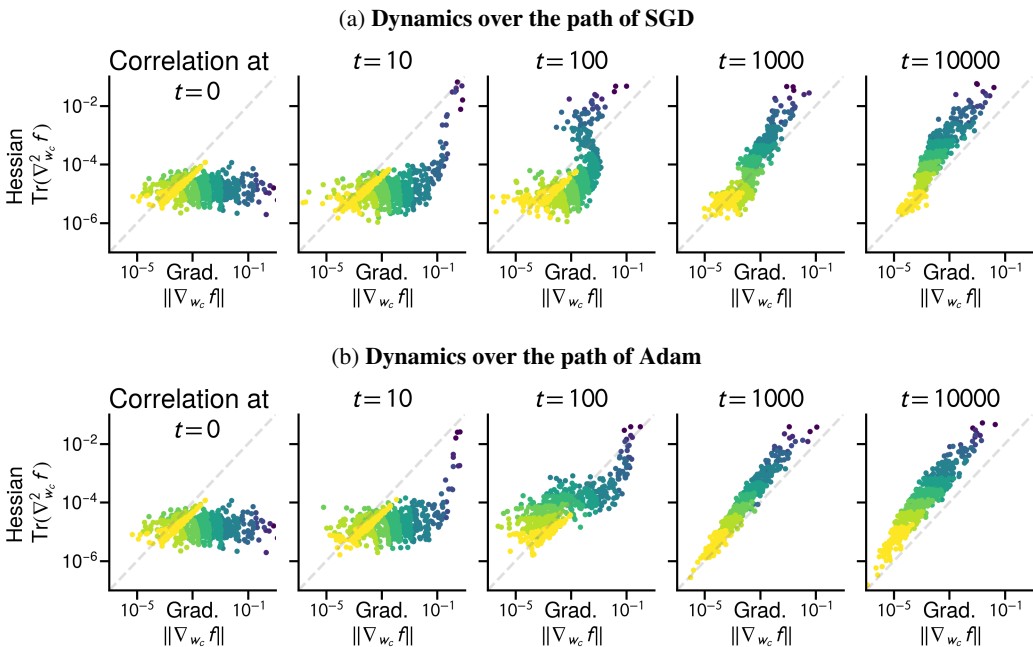

Figure 28: **The gradient-Hessian blocks also become correlated in the last layer of large models.** Reproducing Figure 7 on the GPT2-Small/WikiText-103 problem of Figure 1. Evolution of the gradient norm and Hessian trace for each row $\mathbf{w}_c$ of the last layer throughout optimization, over the path taken by SGD (a) and Adam (b). The color indicates the class frequency, showing that lower (higher) frequency classes have smaller (larger) gradient norm and Hessian trace.

## F.3 CNN on Barcoded+MNIST

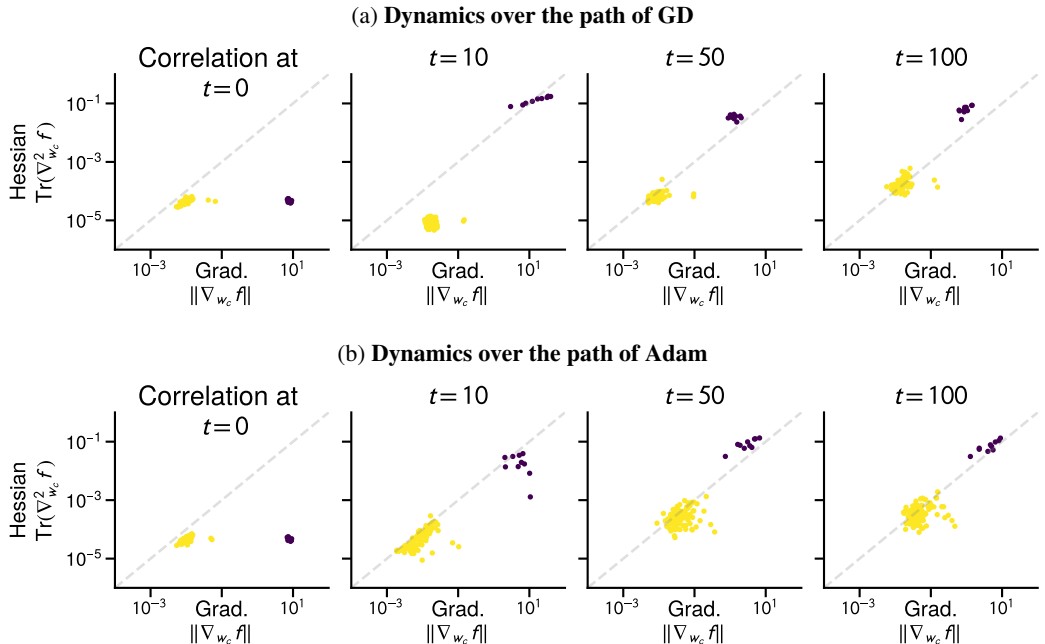

Figure 29: **Evolution of the gradient norm and Hessian trace through optimization.** Taken over the path of GD and Adam on the CNN on imbalanced MNIST in Figure 2. Note that this problem only has two groups of classes with different frequencies; 10 classes have ≈5k samples while 10k classes have 5 samples.

## F.4 ResNet18 on Heavy-Tailed ImageNet

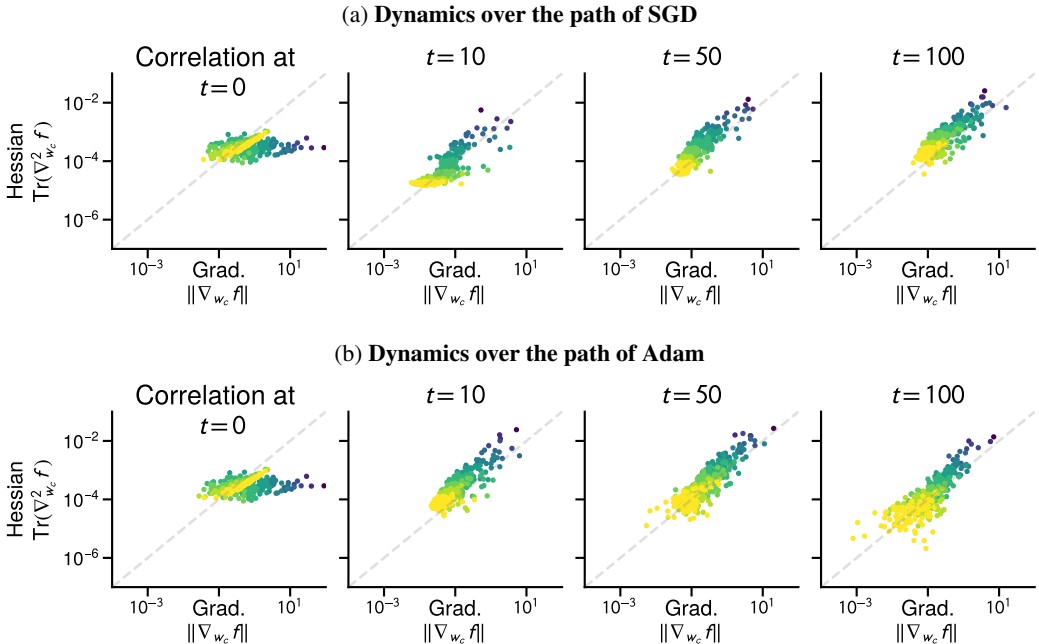

Figure 30: **Evolution of the gradient norm and Hessian trace through optimization.** Taken over the path of SGD and Adam on the ResNet18 on Heavy-Tailed ImageNet in Figure 3.

## F.5 Small Transformer on PTB

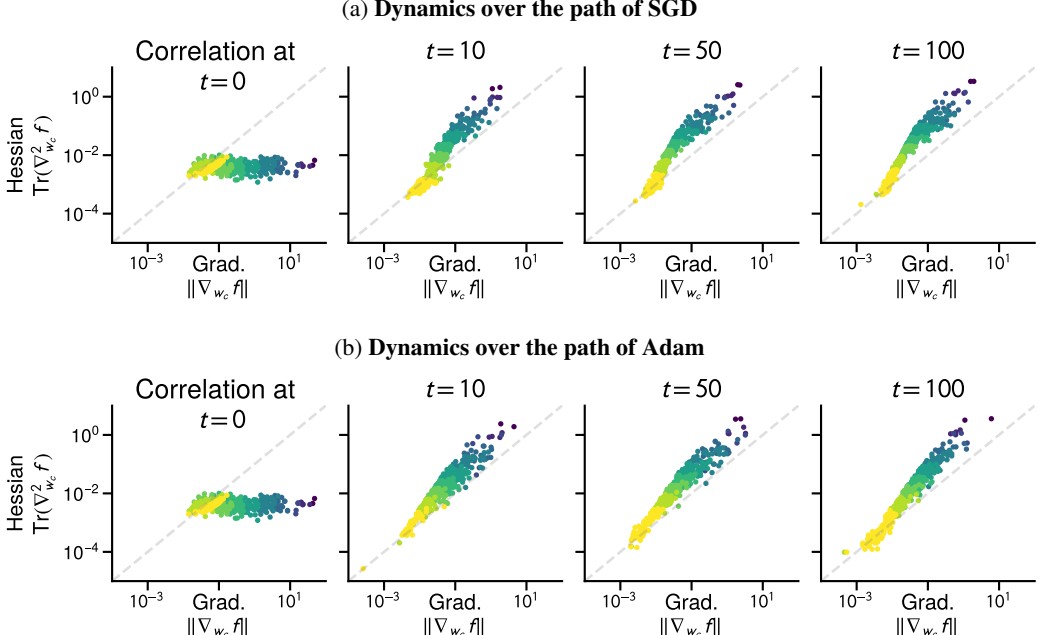

Figure 31: **Evolution of the gradient norm and Hessian trace through optimization.** Taken over the path of SGD and Adam on the small Transformer on PTB in Figure 12.

## F.6 The correlation depends on the path

Proposition 2 requires that the optimizer make progress and assign samples to their correct classes. Indeed, the positive correlation observed in the previous figures is not a global property of the loss function. Not only does it not hold at initialization, where the Hessian is uniform, the correlation can even be reversed in some areas of the parameter space, as shown in Figure 32.

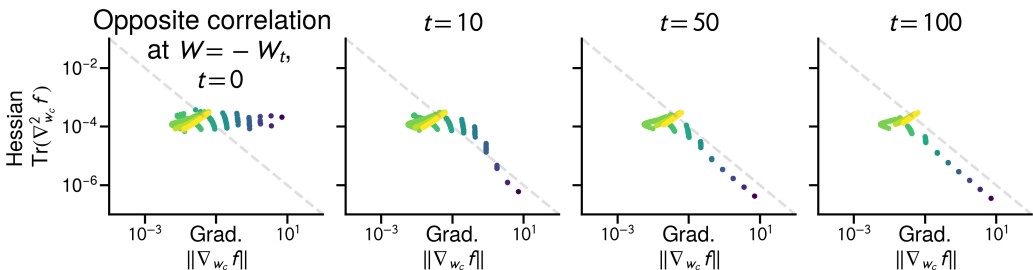

Figure 32: **The correlation only holds while training.** Correlation between the gradient and Hessian blocks through the path $\{-\mathbf{W}_t\}$, where $\mathbf{W}_t$ are the iterates of Adam on the linear model of Figure 4.

# G   Correlation between the gradient and Hessian across blocks

This section gives the proof of Proposition 2 in Section 3.2

**Proposition 2.** *If initialized at $\mathbf{W}_0 = 0$, the gradient and Hessian of the loss $\mathcal{L}$ w.r.t. $\mathbf{w}_k$ are*

$$\nabla_{\mathbf{w}_k} \mathcal{L}(\mathbf{W}_0) = \pi_k \bar{\mathbf{x}}^k - \tfrac{1}{c} \bar{\mathbf{x}}, \qquad\qquad \nabla^2_{\mathbf{w}_k} \mathcal{L}(\mathbf{W}_0) = \tfrac{1}{c}\left(1 - \tfrac{1}{c}\right) \bar{\mathbf{H}}, \qquad\qquad (1)$$

*During training, if the model correctly assigns samples to class $k$ with probability $p$ (Assumption 1),*

$$\begin{aligned} \nabla_{\mathbf{w}_k} \mathcal{L} &= (1-p)\pi_k \, \bar{\mathbf{x}}^k + O\big(\tfrac{1}{c}\big), \\ \nabla^2_{\mathbf{w}_k} \mathcal{L} &= p(1-p)\pi_k \, \bar{\mathbf{H}}^k + O\big(\tfrac{1}{c}\big), \end{aligned} \quad \text{and} \quad \|\nabla_{\mathbf{w}_k} \mathcal{L}\| \sim \left( \frac{1}{p} \frac{\|\bar{\mathbf{x}}^k\|}{\mathrm{Tr}(\bar{\mathbf{H}}^k)} \right) \mathrm{Tr}(\nabla^2_{\mathbf{w}_k} \mathcal{L}) \text{ as } c \to \infty, \quad (2)$$

*for classes where the frequency does not vanish too quickly, $\pi_k = \omega(1/c)$.*

The requirement that the class frequencies do not vanish, $\pi_k = \omega(1/c)$, is necessary to make it possible to discuss class frequencies as $c \to \infty$, unless the class frequencies do not depend on $c$. While the frequencies $\pi_k$ and the number of classes $c$ can be independent, for example if $\pi_k$ follows an exponential decay, $\pi_k \propto 2^{-k}$, it does not hold for all distributions. While it may seem that this result only holds for relatively frequent classes, as it requires $\pi_k c \to \infty$, we can see that nearly all the data comes from classes where this correlation holds when the classes are distributed as $\pi_k \propto 1/k$. Denote by $H(c) = \sum_{k=1}^c 1/k = \Theta(\log c)$. After normalization, we have $\pi_k = 1/kH(c)$. The correlation result holds as long as $\pi_k c \to \infty$, and so it at least holds for the first $k \leq c/\log(c)^2$ classes as $\pi_k c \geq \log(c) \to \infty$. While this only cover a $1/\log(c)^2$ fraction of the classes, those classes account for nearly all the data as

$$\sum_{k=1}^{\lceil \frac{c}{\log(c)} \rceil} \pi_k = \frac{H\big(\lceil c/\log(c)^2 \rceil\big)}{H(c)} = \Theta\left( \frac{\log(c) - 2\log\log(c)}{\log(c)} \right) \to 1.$$

*Proof of Proposition 2.* We first recall the gradient and Hessian for each block $\mathbf{w}_1, ..., \mathbf{w}_c$;

$$\nabla_{\mathbf{w}_k} \ell(\mathbf{W}, \mathbf{x}, \mathbf{y}) = (\mathbf{1}[y = k] - \mathbf{p}(\mathbf{x})_k)\mathbf{x}, \qquad \nabla^2_{\mathbf{w}_k} \ell(\mathbf{W}, \mathbf{x}, \mathbf{y}) = \mathbf{p}(\mathbf{x})_k(1 - \mathbf{p}(\mathbf{x})_k)\mathbf{x}\mathbf{x}^\top,$$

and the definitions of the moments of the data, per class and overall.

$$\bar{\mathbf{x}}^k = \tfrac{1}{n_k} \sum_{i=1:y_i=k}^n \mathbf{x}_i, \quad \bar{\mathbf{x}} = \tfrac{1}{n} \sum_{i=1}^n \mathbf{x}_i, \quad \bar{\mathbf{H}}^k = \tfrac{1}{n_k} \sum_{i=1:y_i=k}^n \mathbf{x}_i\mathbf{x}_i^\top, \quad \bar{\mathbf{H}} = \tfrac{1}{n} \sum_{i=1}^n \mathbf{x}_i\mathbf{x}_i^\top.$$

Our first step is to rewrite the sums for the gradient and Hessian to separate the influence of the samples of the correct class $k$ and the other samples.

$$\begin{aligned} \nabla_{\mathbf{w}_k} \mathcal{L}(\mathbf{W}) &= \frac{1}{n} \sum_{i=1}^n (\mathbf{1}[y_i = k] - \mathbf{p}(\mathbf{x}_i)_k)\mathbf{x}_i, \\ &= \frac{1}{n} \sum_{j=1}^c \sum_{i:y_i=j} (\mathbf{1}[y_i = k] - \mathbf{p}(\mathbf{x}_i)_k)\mathbf{x}_i, & \text{(Split by class)} \\ &= \sum_{j=1}^c \frac{\pi_j}{n_j} \sum_{i:y_i=j} (\mathbf{1}[y_i = k] - \mathbf{p}(\mathbf{x}_i)_k)\mathbf{x}_i, & \text{(Use class frequencies } \pi_j = n_j/n) \\ &= \pi_k \frac{1}{n_k} \sum_{i=1:y_i=k}^n (1 - \mathbf{p}(\mathbf{x}_i)_k)\mathbf{x}_i + \sum_{j=1,j\neq k}^c \frac{\pi_j}{n_j} \sum_{i:y_i=j} (-\mathbf{p}(\mathbf{x}_i)_k)\mathbf{x}_i. \end{aligned}$$

$$\begin{aligned} \nabla^2_{\mathbf{w}_k} \mathcal{L}(\mathbf{W}) &= \frac{1}{n} \sum_{i=1}^n \mathbf{p}(\mathbf{x}_i)_k(1 - \mathbf{p}(\mathbf{x}_i)_k)\mathbf{x}_i\mathbf{x}_i^\top, \\ &= \frac{\pi_k}{n_k} \sum_{i:y_i=k} \mathbf{p}(\mathbf{x}_i)_k(1 - \mathbf{p}(\mathbf{x}_i)_k)\mathbf{x}_i\mathbf{x}_i^\top + \sum_{j=1,j\neq k}^c \frac{\pi_j}{n_j} \sum_{i:y_i=j} \mathbf{p}(\mathbf{x}_i)_k(1 - \mathbf{p}(\mathbf{x}_i)_k)\mathbf{x}_i\mathbf{x}_i^\top. \end{aligned}$$

We can simplify the first terms using the assumption that $p(\mathbf{x}_i)_k = p$ for samples of the correct class,

$$\frac{\pi_k}{n_k} \sum_{i=1:y_i=k}^n (1 - \mathbf{p}(\mathbf{x}_i)_k)\mathbf{x}_i = (1-p)\pi_k\bar{\mathbf{x}}^k, \quad \frac{\pi_k}{n_k} \sum_{i:y_i=k} \mathbf{p}(\mathbf{x}_i)_k(1 - \mathbf{p}(\mathbf{x}_i)_k)\mathbf{x}_i\mathbf{x}_i^\top = p(1-p)\pi_k\bar{\mathbf{H}}^k.$$

We introduce the following shorthands for the second terms,

$$\mathbf{d}_k = c \sum_{j=1, j\neq k}^{c} \frac{\pi_j}{n_j} \sum_{i:y_i=j} (-\mathbf{p}(\mathbf{x}_i)_k)\mathbf{x}_i, \quad \mathbf{D}_k = c \sum_{j\neq k} \frac{\pi_j}{n_j} \sum_{i:y_i=j} \mathbf{p}(\mathbf{x}_i)_k(1-\mathbf{p}(\mathbf{x}_i)_k)\mathbf{x}_i\mathbf{x}_i^\top.$$

Using those simplifications, we obtain that

$$\nabla_{\mathbf{w}_k}\mathcal{L}(\mathbf{W}) = (1-p)\pi_k\bar{\mathbf{x}}^k + \frac{1}{c}\mathbf{d}_k, \qquad \nabla^2_{\mathbf{w}_k}\mathcal{L}(\mathbf{W}) = p(1-p)\pi_k\bar{\mathbf{H}}^k + \frac{1}{c}\mathbf{D}_k.$$

The terms $\mathbf{d}_k$, $\mathbf{D}_k$ are averages of terms weighted by $c\mathbf{p}(\mathbf{x}_i)_k$, which by assumption is $O(1)$, and as such both $\|\mathbf{d}_k\|$ and $\mathrm{Tr}(\mathbf{D}_k)$ are $O(1)$. The ratio between the two will be dominated by the contribution of their first term as long as $\pi_k$ dominates $1/c$, in the sense that $\lim_{c\to\infty} \frac{1}{\pi_k c} \to 0$, as

$$\lim_{c\to\infty} \frac{\|\nabla_{\mathbf{w}_k}\mathcal{L}\|}{\mathrm{Tr}(\nabla^2_{\mathbf{w}_k}\mathcal{L})} = \lim_{c\to\infty} \frac{\left\|(1-p)\pi_k\bar{\mathbf{x}}^k + \frac{1}{c}\mathbf{d}_k\right\|}{\mathrm{Tr}(p(1-p)\pi_k\bar{\mathbf{H}}^k + \frac{1}{c}\mathbf{D}_k)}$$

$$= \lim_{c\to\infty} \frac{\left\|(1-p)\bar{\mathbf{x}}^k + \frac{1}{c\pi_k}\mathbf{d}_k\right\|}{\mathrm{Tr}(p(1-p)\pi_k\bar{\mathbf{H}}^k + \frac{1}{c\pi_k}\mathbf{D}_k)} = \frac{1}{p}\frac{\|\bar{\mathbf{x}}^k\|}{\mathrm{Tr}(\bar{\mathbf{H}}^k)}. \qquad \square$$

### G.1 Off-diagonal blocks are orders of magnitude smaller than diagonal blocks

Our discussion Section 3.2 ignored the impact of off-diagonal blocks. In this section, we show that they are small. The diagonal and off-diagonal blocks of the matrix for $k \neq k'$ is

$$\begin{aligned} \mathbf{H}_{kk} &:= \nabla^2_{\mathbf{w}_k}\ell(\mathbf{W}, \mathbf{x}, y) &&= \mathbf{p}(\mathbf{x})_k(1 - \mathbf{p}(\mathbf{x})_k)\mathbf{x}\mathbf{x}^\top, \\ \text{and for } j \neq k, \quad \mathbf{H}_{kj} &:= \nabla_{\mathbf{w}_k}\nabla_{\mathbf{w}_{k'}}\ell(\mathbf{W}, \mathbf{x}, \mathbf{y}) &&= \mathbf{p}(\mathbf{x})_k(-\mathbf{p}(\mathbf{x})_{k'})\mathbf{x}\mathbf{x}^\top. \end{aligned}$$

From this, we can see that, on average, the magnitude of the off-diagonal blocks will be smaller than that of the diagonal blocks, as

$$\mathbf{H}_{kk} = -\sum_{j=1, j\neq k}^{c} \mathbf{H}_{kj},$$

because $\sum_{k'=1, k'\neq k}^{c} \mathbf{p}(\mathbf{x})_k\mathbf{p}(\mathbf{x})_{k'} = \mathbf{p}(\mathbf{x})_k(1 - \mathbf{p}(\mathbf{x})_k)$, This means that the matrix $\mathbf{T} : [c \times c]$ formed by taking the trace of the blocks, $\mathbf{T}_{jk} = \mathrm{Tr}(\mathbf{H}_{jk})$, is diagonally dominant.

Figures 9 and 33 show that the magnitude of the entries of the Hessian in off-diagonal blocks is orders of magnitude smaller than those of the diagonal blocks. Instead of plotting the $[cd \times cd]$ Hessian, we subsample 40 classes and 40 input dimensions and plot the resulting $[160 \times 160]$ entries at different points throughout the trajectory of Adam on the problem of Figure 4. Figure 9 shows the matrices with classes sampled uniformly and Figure 33 with classes sampled log-uniformly

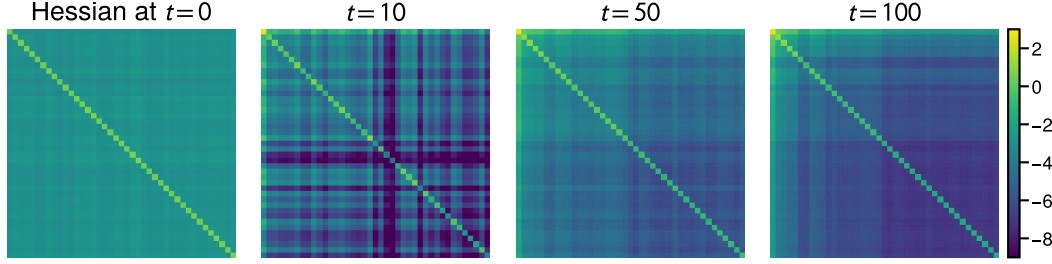

Figure 33: **The off-diagonal blocks are much smaller than the diagonal blocks.** Showing the magnitude $\log_{10}(|(\nabla^2\mathcal{L})_{ij}|)$ for a $[160 \times 160]$ subset of the Hessian, sampling 40 classes and 40 input dimensions uniformly.

# H Continuous time GD and sign descent on a simple imbalanced problem

We give the proof of Theorem 3 on the simple imbalanced setting, restated here for convenience.

**Simple imbalanced setting.** *Consider $c$ classes with frequencies $\pi_1, ..., \pi_c$ where all samples from a class are the same, $\mathbf{x}_i = \mathbf{e}_k$ if $y_i = k$, where $\mathbf{e}_k$ is the $k$th standard basis vector in $\mathbb{R}^c$.*

**Theorem 3.** *On the* simple imbalanced setting*, gradient flow and continuous time sign descent initialized at $\mathbf{W} = 0$ minimize the loss of class $k$, $\ell_k(t) = -\log(\sigma(\mathbf{W}(t)\mathbf{e}_k)_k)$, at the rate*

$$\text{Gradient flow:} \quad \ell_k(t) = \Theta(1/\pi_k t), \quad \text{Continuous time sign descent:} \quad \ell_k(t) = \Theta(e^{-ct}).$$

We separate the proof for gradient flow into 3 parts. Lemma 4 simplifies the dynamics into smaller, independent differential equations, Lemma 5 solves the differential equation and Lemma 6 bounds the loss. The proof uses similar tools as for the gradient flow dynamics studied by Cabannes et al. (2024), but we focus instead on the loss per class. We treat continuous time sign descent separately in Lemma 7.

**Notation.** If $\mathbf{W}$ is a $[a \times b]$ matrix, then $\mathbf{w}_1, ..., \mathbf{w}_a$ are the rows and $\mathbf{w}^1, ..., \mathbf{w}^b$ are the vectors, and $w_{ij}$ is the entry at the $i$th column, $j$th row. For brevity, we use $z = c - 1$ as the term appears often.

**Lemma 4** (Separation of the dynamics). *The dynamics of the parameter matrix $\mathbf{W}$ separate into $c$ 2-dimensional differential equations, $w_{kk}(t) = a_k(t)$ and $w_{jk}(t) = b_k(t)$ for $j \neq k$, where*

$$a_k(0) = 0, \qquad \frac{\mathrm{d}}{\mathrm{d}t}a_k = \pi_k\left(1 - \frac{\exp(a_k)}{\exp(a_k) + (c-1)\exp(b_k)}\right),$$

$$b_k(0) = 0, \qquad \frac{\mathrm{d}}{\mathrm{d}t}b_k = \pi_k\left(\quad - \frac{\exp(b_k)}{\exp(a_k) + (c-1)\exp(b_k)}\right).$$

*Proof.* Our goal is to simplify the dynamics starting at $\mathbf{W}(0) = 0$ and following the gradient flow $\frac{\mathrm{d}}{\mathrm{d}t}\mathbf{W} = -\nabla\mathcal{L}(\mathbf{W})$, where $\mathbf{W} : [c \times d]$. For the simplified setting, we have that $d = c$ are the inputs are the standard basis vectors in $\mathbb{R}^c$. The derivative of $\mathcal{L}$ w.r.t. a single element $w_{kj}$ is

$$\partial_{w_{kj}}\mathcal{L}(\mathbf{W}) = -\pi_k\mathbf{1}[k = j] + \pi_j\sigma(\mathbf{w}^j)_k.$$

As $\partial_{w_{kj}}$ only depends on $\mathbf{w}^j$ for all $k$, The dynamics are independent across the columns of $\mathbf{W}$, giving $c$ independent equations in $\mathbb{R}^c$,

$$\mathbf{w}^j(0) = 0, \qquad \frac{\mathrm{d}}{\mathrm{d}t}\mathbf{w}^j = \pi_j(\mathbf{e}_j - \sigma(\mathbf{w}^j)).$$

To further simplify the dynamics, we use the fact that the weights that are not associated with the correct class have the same dynamics. For any indices $i, j$ different from $k$, $w_{ik}(t) = w_{jk}(t)$. They have the same derivatives if they have the same value, as

$$-\frac{\mathrm{d}}{\mathrm{d}t}w_{ik} = \pi_k\sigma(\mathbf{w}^k)_i = \pi_k\frac{\exp(w_{ik})}{\sum_{k'}\exp(w_{k'k})} = \pi_k\frac{\exp(w_{jk})}{\sum_{k'}\exp(w_{k'k})} = \pi_k\sigma(\mathbf{w}^k)_j = -\frac{\mathrm{d}}{\mathrm{d}t}w_{jk},$$

so they will have the same dynamics and the equation can be reduced to a system of 2 variables, $w_{kk} = a_k$ and $w_{jk} = b_k$ for any $j \neq k$, with

$$a_k(0) = 0, \qquad \frac{\mathrm{d}}{\mathrm{d}t}a_k = \pi_k\left(1 - \frac{\exp(a_k)}{\exp(a_k) + (c-1)\exp(b_k)}\right),$$

$$b_k(0) = 0, \qquad \frac{\mathrm{d}}{\mathrm{d}t}b_k = \pi_k\left(\quad - \frac{\exp(b_k)}{\exp(a_k) + (c-1)\exp(b_k)}\right). \qquad \square$$

**Lemma 5** (Solution of the dynamics). *For a given class with frequency $\pi$, the dynamics of the parameters $a$ and $b$ in Lemma 4 evolve as follows, using the shortcuts $f(t) = 1 + c\pi t$ and $z = c - 1$,*

$$a(t) = \frac{1}{c}\left(f(t) - zW\left(\frac{1}{z}\exp\left(\frac{1}{z}f(t)\right)\right)\right) \qquad\qquad b(t) = -\frac{1}{z}a(t),$$

*Proof.* We want the solution to the differential equation

$$a(0) = 0 \qquad\qquad \frac{\mathrm{d}}{\mathrm{d}t}a = \pi\left(1 - \frac{\exp(a)}{\exp(a) + (c-1)\exp(b)}\right),$$

$$b(0) = 0 \qquad\qquad \frac{\mathrm{d}}{\mathrm{d}t}b = \pi\left( - \frac{\exp(b)}{\exp(a) + (c-1)\exp(b)}\right).$$

The general solution, ignoring the initial conditions, uses the Lambert $W$ function and constants $K_1, K_2$.[1] For brevity, we introduce the shortcut $z = c - 1$.

$$a(t) = \frac{1}{zc}\left(ce^{-K_1}K_2 + cz\pi t - z^2 W\left(\frac{1}{z}\exp\left(\frac{c}{z^2}(z\pi t + e^{-K_1}K_2) - K_1\right)\right)\right),$$

$$b(t) = K_1 - \frac{1}{z^2 c}\left(ce^{-K_1}K_2 + cz\pi t - z^2 W\left(\frac{1}{z}\exp\left(\frac{c}{z^2}(z\pi t + e^{-K_1}K_2) - K_1\right)\right)\right).$$

We need to set $K_1, K_2$ to satisfy the initial conditions $a(0) = b(0) = 0$. As $b(t) = K_1 - a(t)/z$, we must have that $K_1 = 0$, giving the simplification

$$a(t) = \frac{1}{zc}\left(cK_2 + cz\pi t - z^2 W\left(\frac{1}{z}\exp\left(\frac{c}{z^2}(z\pi t + K_2) - K_1\right)\right)\right), \qquad b(t) = -\frac{1}{z}a(t).$$

To set $K_2$, we need to have

$$0 = zca(0) = cK_2 - z^2 W\left(\frac{1}{z}\exp\left(K_2\frac{c}{z^2}\right)\right) \implies W\left(\frac{1}{z}\exp\left(K_2\frac{c}{z^2}\right)\right) = \frac{c}{z^2}K_2$$

Since $W(xe^x) = x$ for $x > 0$, the equation is satisfied for $K_2 = \frac{z}{c}$, as we get $W\left(\frac{1}{z}e^{\frac{1}{z}}\right) = \frac{1}{z}$, giving

$$a(t) = \frac{1}{c}\left(1 + c\pi t - zW\left(\frac{1}{z}\exp\left(\frac{1}{z}(1 + c\pi t)\right)\right)\right) \qquad\qquad b(t) = -\frac{1}{z}a(t). \qquad \square$$

**Lemma 6** (Bound for the loss). *For $t$ sufficiently large such that $1 + c\pi_k t \geq z\log z + 1$,*

$$\ell_k(t) = \Theta\left(\frac{1}{\pi_k t}\right).$$

Using the simplification derived in Lemma 4 and the solution of the differential equation in Lemma 5, we can rewrite the loss for a specific class as a function of time as

$$L_k(\mathbf{W}) := -\log(\sigma(\mathbf{W}\mathbf{e}_k)_k) = -\log\left(\frac{\exp(w_{kk})}{\sum_{j=1}^{c}\exp(w_{jk})}\right),$$

$$\ell_k(t) := L_k(\mathbf{W}(t)) = -\log\left(\frac{\exp(a_k(t))}{\exp(a_k(t)) + (c-1)\exp(b_k(t))}\right) = \log(1 + (c-1)\exp(cb_k(t))),$$

where the equality uses that $a_k(t) = (c-1)b_k(t)$. For brevity, we will drop the index $k$ in $a_k$, $b_k$, $\ell_k$ and $\pi_k$ and use the shortcut $z = c - 1$, bounding the quantity

$$\ell(t) = \log(1 + z\exp(cb(t))).$$

Expanding the definition of $b(t)$ using Lemma 5, we have

$$z\exp(cb(t)) = z\exp\left(-\frac{1}{z}\left(f(t) - zW\left(\frac{1}{z}\exp\left(\frac{1}{z}f(t)\right)\right)\right)\right), \qquad \text{where} \qquad f(t) = 1 + c\pi t.$$

To simplify the $W$ function, we use the fact that for $x > e$ (Hoorfar and Hassani, 2008, Theorem 2.7)

$$W(x) = \log(x) - \log(\log(x)) + \delta(x) \qquad \text{where} \qquad \frac{1}{2} \leq \delta(x)\frac{\log(x)}{\log(\log(x))} \leq \frac{e}{e-1}.$$

---

[1] WolframAlpha solution for $\pi = 1$: https://www.wolframalpha.com/input?i=d/dt+x(t)+=+1-exp(x(t))/(exp(x(t))+c*exp(y(t))),+d/dt+y(t)+=+-exp(y(t))/(exp(x(t))+c*exp(y(t)))

To use this bound on $W\left(\frac{1}{z}\exp\left(\frac{1}{z}f(t)\right)\right)$, we need $\frac{1}{z}\exp\left(\frac{1}{z}f(t)\right) \geq e$, which is satisfied for $t$ sufficiently large, once $f(t) \geq z(\log z + 1)$.

Using that $\log\left(\frac{1}{z}\exp\left(\frac{1}{z}f(t)\right)\right) = \frac{1}{z}f(t) - \log(z)$, and writing $h(t) = \delta\left(\frac{1}{z}\exp\left(\frac{1}{z}f(t)\right)\right)$, we have

$$f(t) - zW\left(\frac{1}{z}\exp\left(\frac{1}{z}f(t)\right)\right) = f(t) - z\left(\frac{1}{z}f(t) - \log(z) - \log\left(\frac{1}{z}f(t) - \log(z)\right) + h(t)\right),$$
$$= z(\log(f(t) - z\log(z)) - h(t)),$$

giving the simplification

$$z\exp(cb(t)) = z\exp\left(-\frac{1}{z}\left(f(t) - zW\left(\frac{1}{z}\exp\left(\frac{1}{z}f(t)\right)\right)\right)\right),$$
$$= z\exp(-\log(f(t) - z\log(z)) + h(t)) = \frac{z\exp(h(t))}{f(t) - z\log z},$$

This gives the average loss

$$\ell(t) = \log(1 + z\exp(cb(t))) = \log\left(1 + \frac{z\exp(h(t))}{f(t) - z\log z}\right)$$

To bound this expression, we can use that $\frac{z\exp(h(t))}{f(t) - z\log z} \geq 0$ after $f(t) \geq z\log z$, which we have already assumed to apply the bound on the $W$ function, and use the bounds $\frac{x}{1+x} \leq \log(1+x) \leq x$ to get

$$\frac{z\exp(h(t))}{f(t) - z\log z + z\exp(h(t))} \leq \ell(t) \leq \frac{z\exp(h(t))}{f(t) - z\log z}.$$

As $h(t)$ is upper bounded by a constant and $\lim_{t\to\infty} h(t) = 0$, $\lim_{t\to\infty} \exp(h(t)) = 1$, we have

$$\ell(t) = \Theta\left(\frac{z}{f(t) - z\log z}\right) = \Theta\left(\frac{1}{\pi t}\right).$$

**Lemma 7.** *The loss at time $t$ for continuous time sign descent is $\ell_k(t) = \log(1 + (c-1)\exp(-ct))$*

*Proof.* The same decomposition as in Lemma 4 hold, with the dynamics

$$a_k(0) = 0, \quad \frac{\mathrm{d}}{\mathrm{d}t}a_k = 1, \quad a_k(t) = t, \qquad b_k(0) = 0, \quad \frac{\mathrm{d}}{\mathrm{d}t}b_k = -1, \quad b_k(t) = -t,$$

leading to the following loss

$$\ell_k(t) = \log(1 + (c-1)\exp(-ct)) = \Theta(z\exp(-ct)). \qquad \square$$

