# OpenReview forum: "Heavy-Tailed Class Imbalance and Why Adam Outperforms Gradient Descent on Language Models"
_NeurIPS.cc/2024/Conference — NeurIPS 2024 spotlight_

### Official Review · Reviewer_CdGj · 2024-07-09

**Soundness:** 4
**Presentation:** 4
**Contribution:** 4
**Rating:** 8
**Confidence:** 4

**Summary:**

This paper addresses the important problem of why Adam outperforms SGD for language tasks, proposing that heavy-tailed class imbalance in the training dataset is the key factor in this performance gap. A series of experiments demonstrate that Adam consistently outperforms SGD under heavy-tailed class imbalance because SGD shows slow or no progress on low-frequency classes, whereas Adam progresses independently of class frequencies. This finding holds true across different data types (language and image data), architectures (CNNs and Transformers), and the stochasticity of the training algorithm (mini-batch and full-batch).

The paper also theoretically investigates a linear softmax classification problem, proving that the convergence of gradient flow (as a proxy for GD) depends heavily on class frequencies, whereas the convergence of a continuous-time variant of sign descent (as a proxy for Adam) is independent of class frequencies. This theoretical result establishes the benefits of sign-based methods under heavy-tailed class imbalance for a linear model.

**Strengths:**

- The finding that heavy-tailed class imbalance leads to the performance gap between Adam and SGD is a solid and novel contribution. This offers new insights into understanding the important question of why and when Adam outperforms SGD.
- The experiments are well-designed and robust, including thorough ablation studies that strongly support the hypothesis.
- I think it's great that the authors can rigorously establish the effect of class frequencies on the training speed of GD and sign descent in a simplified setting. Additionally, the investigation into the correlation between gradient and Hessian across coordinates provides valuable insights into the underlying mechanisms by which sign-based methods benefit from heavy-tailed class imbalance.

**Weaknesses:**

- The linear model studied is an oversimplified setting and is designed to be biased towards sign descent. Therefore, it is unclear whether the insights gained from this model extend to practical settings, particularly in understanding why Adam benefits from a heavy-tailed class imbalance in real-world scenarios. Specifically, it remains uncertain whether the correlation between gradient, Hessian, and class frequencies observed in the linear model holds true in more complex, practical settings.

**Questions:**

- In the gradient norm and Hessian trace plots in Figures 24-26 in Appendix G, how are the weight blocks $w_c$ defined? Do they correspond to the last layer of each architecture? If so, how does class imbalance affect other layers beyond the last layer?
- Could the authors please cite the relevant works by Zhang et al. (https://arxiv.org/abs/2402.16788) and Xie and Li (https://arxiv.org/abs/2404.04454) that also study the benefits of Adam? In particular, the block heterogeneity discussed by Zhang et al. seems closely related to the intuition in this paper. Additionally, Xie and Li observe that Adam outperforms GD when the loss function has better properties under $\ell_\infty$ geometry, which could relate to heavy-tailed class imbalance. Therefore, I wonder if the authors can discuss and reconcile these works with the findings on heavy-tailed class imbalance.

**Limitations:**

The paper discusses its limitations and future directions in detail. For additional potential limitations, please refer to the 'Weakness' section.

---

> ### Author Rebuttal · Authors · 2024-08-05
>
> **Weight blocks for the gradient-Hessian correlation plots**
> > In the gradient norm and Hessian trace plots in Figures 24-26 in Appendix G, how are the weight blocks $w_c$ defined? Do they correspond to the last layer of each architecture?
>
> This is correct. The plots show the last layer for each problem. The last layer is a $[D \times C]$ matrix, where $D$ is the dimension of the last hidden layer and $C$ is the number of classes, and each $w_c$ correspond to a column of this matrix. We will make it clear in Appendix G.
>
> **Effect beyond last layer**
> > If so, how does class imbalance affect other layers beyond the last layer?
>
> This question is difficult to answer as there is no direct "mapping" between specific weights and classes beyond the last layer. It is likely that the slow performance of GD on the last layer leads to slower performance on earlier layers. For example, it might be that some feature relevant to a particular class can only really be learned once this class is sufficiently separated from the others. But formalizing this idea is far from trivial.
>
> However, other ablation studies that appeared after our submission also provide evidence that one of the primary benefit of Adam over GD is to fix the effect of class imbalance in the last layer. We discuss those works below.
>
> **Correlation appears on realistic models**
> > The linear model studied is an oversimplified setting and is designed to be biased towards sign descent. Therefore, it is unclear whether the insights gained from this model extend to practical settings, particularly in understanding why Adam benefits from a heavy-tailed class imbalance in real-world scenarios. Specifically, it remains uncertain whether the correlation between gradient, Hessian, and class frequencies observed in the linear model holds true in more complex, practical settings.
>
> While we agree that the toy model of §3.3 is not a realistic depiction of modern complex models, we do believe that the main take-away message, that the performance of GD suffers in heavy-tailed class imbalance, holds across models. Proposition 2 only describes a correlation between gradients and Hessians for rows of the last layer, but the performance gap between GD and Adam already appears when tuning only the last layer, keeping everything else frozen (see Figure 5). The effect described by Proposition 2 should still hold for complex models (assuming the inputs to the last layer do not change drastically to undo the effect of the scaling by the assignment property) and we observe this correlation beyond the linear model. Appendix G shows this effect on a small transformer on PTB, a CNN on the imbalanced MNIST dataset and a ResNet18 on the imbalanced ImageNet dataset.
>
> We include an additional plot in [**the supplementary pdf for the response**](https://openreview.net/attachment?id=xsoeNVdrpL&name=pdf) which shows that the correlation between gradients and Hessian observed in Fig 7 also holds on GPT2/WikiText, using the same training procedure as Fig. 1.
>
> **Related work**
> > Could the authors please cite the relevant works by [Zhang et al.](https://arxiv.org/abs/2402.16788) and [Xie and Li](https://arxiv.org/abs/2404.04454) that also study the benefits of Adam?
>
> We will include a discussion of those works. Follow-up works to [Zhang et al. (2024a)](https://arxiv.org/abs/2402.16788) from [Zhang et al. (2024b)](https://arxiv.org/abs/2406.16793) and [Zhao et al (2024)](https://arxiv.org/abs/2407.07972), (both of which where arXived after our submission) also give some insights on the benefits of Adam beyond the last layer, and we will include a discussion of those works.
>
> In addition to their work showing that the Hessian of transformers has a block-diagonal structure ([Zhang et al., 2024a](https://arxiv.org/abs/2402.16788)), [Zhang et al. (2024b)](https://arxiv.org/abs/2406.16793) show in a follow-up ablation studies. Instead of looking for properties of the model, they study changes to the training procedure and show that the diagonal/sign-descent behavior of Adam is mostly redundant. Revisiting earlier layer-wise normalization approaches, they show empirically that most of the benefit of Adam can be recovered by using a learning rate per weight matrix rather than per coordinate (dividing by the average of the gradient magnitudes). They show that per-layer normalization achieves the same performance as Adam, as long as this scheme is not applied to the last layer. [Zhao et al (2024)](https://arxiv.org/abs/2407.07972) provide a similar ablation study, but further shows that most layers can be trained using unnormalized SGD updates, except for LayerNorm layers and the last layer.
>
> Both the experiments of [Zhang et al. (2024b)](https://arxiv.org/abs/2406.16793) and [Zhao et al (2024)](https://arxiv.org/abs/2407.07972)  indicate that the performance suffers unless the last layer is normalized independently for each class. Although neither paper specifically shows that this effect is due to class imbalance, their observations support our result that the primary benefit of Adam over SGD comes from the per-class normalization of the last layer to address class imbalance, and show that the impact of choosing an SGD or Adam-style update on the rest of the network has a smaller impact.
>
> ---
>
> **References**
> - [**Zhang et al. (2024a)**](https://arxiv.org/abs/2402.16788)
>   Y. Zhang, C. Chen, T. Ding, Z. Li, R. Sun, Z. Luo
>   Why Transformers Need Adam: A Hessian Perspective
>   https://arxiv.org/abs/2402.16788
> - [**Zhang et al (2024b)**](https://arxiv.org/abs/2406.16793)
>   Y. Zhang, C. Chen, Z. Li, T. Ding, C. Wu, Y. Ye, Z. Luo, R. Sun
>   Adam-mini: Use Fewer Learning Rates To Gain More
>   https://arxiv.org/abs/2406.16793
> - [**Zhao et al (2024)**](https://arxiv.org/abs/2407.07972)
>   R. Zhao, D. Morwani, D. Brandfonbrener, N. Vyas, S. Kakade
>   Deconstructing What Makes a Good Optimizer for Language Models
>   https://arxiv.org/abs/2407.07972

---

> > ### Comment · Reviewer_CdGj · 2024-08-08
> >
> > Thank you for your detailed clarifications and additional ablation studies. I have no further questions. I appreciate the authors' efforts and am happy to increase my review score, voting for strong acceptance. I look forward to seeing the additional experiments and discussions incorporated into the revision.

---

### Official Review · Reviewer_LhRX · 2024-07-11

**Soundness:** 4
**Presentation:** 3
**Contribution:** 3
**Rating:** 7
**Confidence:** 4

**Summary:**

This paper argues that heavy-tailed class imbalances in natural language datasets, which follow because some words are much more frequent than others, causes (or significantly underlies) the performance gap typically observed between Adam and SGD. To make this argument, the authors
1. reproduce the performance gap by training different language models on some standard datasets, like WikiText-103 and PTB;
2. empirically demonstrate that the gap between Adam and SGD can also be reproduced in vision tasks, in which it’s normally absent, by training common vision models on artificially created image datasets with heavy-tailed class imbalances;
3. introduce a simple linear model trained on synthetic uniform data, whose class frequencies follow a power law, and again reproduce the performance gap;
4. rule out batch noise as the reason underlying the performance gap, by demonstrating that the latter remains even when they train their models with full (i.e. deterministic) gradient descent; and
5. demonstrate that the performance gap can be reduced by upweighting the loss of low-frequency classes.

Having done these experimental investigations, the authors then attempt to provide some understanding and intuition as to why these class imbalances affect SGD. The authors study a collection of simplified models and training problem and argue that the class imbalance leads to correlations between gradients, Hessians and class probabilities, which appear to help Adam during training.

**Strengths:**

- This paper studies an important problem in modern machine learning, viz. the performance gap of common training algorithms, specially when dealing with language models.
- The problem and hypothesis are very clearly stated.
- The paper is very well written and structured. Indeed, the authors first present their empirical evidence and only later attempt to provide some intuitive understanding of the phenomenon under investigation.
- As summarised above, the paper provides significant empirical evidence that supports their main hypothesis, namely that heavy-tailed class imbalances in the training datasets cause the performance gap between Adam and SGD.
- The paper also provides some theoreticall understanding as to why the performance gaps actually takes place.

In sum, this paper is very sound, tackles a relevant problem and, in my view, represents an important contribution to the machine learning community.

**Weaknesses:**

The main weaknesses or limitations of the paper pertain to the simplified models and assumptions in the theoretical investigations. However, the authors explicitly acknowledge and discuss them.

Although the paper is very well written and structured, the following suggestions might improve readability:
- The authors could explicitly mention that they reproduce the performance gap between Adam and SGD in language tasks, for different datasets and models. This information is hidden in Appendix A and could escape the casual reader.
- Some captions in the figures are misleading. The authors perform experiments both with SGD and full GD and the way these are labelled, or referred to, in some captions is confusing. For example, the caption in figure 2 refers to SGD while the figure contains results for GD. Similarly the caption in figure 1 refers to GD but the figure contains results for SGD.
- Perhaps section 2.2 could be moved inside section 3?

Typos: lines 152, 160 and 225.

**Questions:**

1. Do you know why SGD eventually finds the minimum in the imbalanced MNIST case but does not in the imbalanced ImageNet?

**Limitations:**

Yes, they authors addressed the limitations of their study.

---

> ### Author Rebuttal · Authors · 2024-08-05
>
> **Clarification**
> > Do you know why SGD eventually finds the minimum in the imbalanced MNIST case but does not in the imbalanced ImageNet?
>
> We think MNIST and ImageNet might have been flipped in this comment. GD and Adam are closer at the end of the given budget on imbalanced ImageNet (Fig. 3) than on on imbalanced MNIST (Fig. 2), where GD appears to stall. We assume the intended phrasing was as given below and will comment on this. Please let us know if we misunderstood your question.
>
> > why SGD [does not] find the minimum in the imbalanced MNIST case but [eventually does] in the imbalanced ImageNet
>
> **Does GD get stuck on Imbalanced MNIST (Fig. 2)**
> GD does eventually find a good solution for MNIST, if run for much longer than Adam. See the longer training run in [**the supplementary pdf for the response**](https://openreview.net/attachment?id=xsoeNVdrpL&name=pdf).
>
> The apparent "plateau" reached by GD is not due to being stuck in a local minimum, but rather due to the optimization progress being slow compared to Adam. This issue is less apparent on ImageNet, possibly because the problem is more complex and Adam cannot find a good solution in 100 steps (note that Fig. 2 (MNIST) shows 300 iterations while Fig. 3 (ImageNet) shows 1500).
>
> We comment on this behaviour in Appendix C.2, showing an example on a linear model where GD appears to stall on a short horizon (100 steps) but converges if run for longer (10k steps). As this question is likely to come up for more readers, we will mention this point explicitly in the MNIST paragraph (L96-103), add a forward reference to §C.2, and add the longer MNIST run to the appendix.
>
> **Inconsistency in references to SGD or GD between caption/figures**
> Those are indeed typos, apologies for the confusion. We will double-check those references.
>
> **Remaining comments**
> We will fix the typos and add references to the appendix to highlight the results on other language tasks.

---

> > ### Comment · Reviewer_LhRX · 2024-08-09
> >
> > Yes, I flipped MNIST with ImageNet, my mistake. I thank the authors for their rebuttal and maintain my score.

---

### Official Review · Reviewer_HWpb · 2024-07-12

**Soundness:** 3
**Presentation:** 3
**Contribution:** 3
**Rating:** 7
**Confidence:** 4

**Summary:**

The paper shows that under the class imbalance setting which is natural in language tasks, Adam can be faster than SGD. Meanwhile, the authors investigate the linear model deeply showing the relationship between gradient and Hessian and the convergence speed of sign-gd and gd algorithm.

**Strengths:**

1. The authors explain that class imbalance is a reason that Adam can outperform SGD.

2. For linear models, the authors establish the relationship between hessian and gradient showing the "correctness" of Adam who approximates Lipschitz with gradients.

3. Further, the authors point out that sign-gd can converge faster than gd for linear models.

**Weaknesses:**

1. All of the results are based on that the parameters of different classes are separable.

2. Since the optimal solution of NN can not be infinity due to some generalization constraints (e.g. adding weight decay), will the sign algorithm still be faster than gd?

**Questions:**

When the parameters are not separable, does the conclusion still hold?

---

> ### Author Rebuttal · Authors · 2024-08-05
>
> **Does the conclusion still hold when the parameters are not separable**
> > When the parameters are not separable, does the conclusion still hold?
> > Since the optimal solution of NN can not be infinity due to some generalization constraints (e.g. adding weight decay), will the sign algorithm still be faster than gd?
>
> We assume you have in mind phenomena such as the one identified by Nacson et al. (2019), where GD can be slower than normalized methods to converge (in direction) as the gradients and Hessian goes to 0. While our experiments use expressive models that can reach good classification accuracy on all classes, the observed behaviour does not require separability nor that the weights go to $\infty$. The behaviour we identify is different, and appears at the start of training, well before the gradients and Hessian go to 0.
>
> In [**the supplementary pdf for the response**](https://openreview.net/attachment?id=xsoeNVdrpL&name=pdf), we show that the distinction between Adam and GD still appears when the model is regularized. The plots show the dynamics of the softmax loss (without the regularization) on a linear softmax model on the small Random Heavy-Tailed Labels dataset with varying levels of $L_2$ regularization ($\lambda \in [10^{-4}, 10^{-3}, 10^{-2}, 10^{-1}]$). For small values of $\lambda$, the weights of the model do not diverge to $\infty$ but the training dynamics are very similar to the unregularized case. Of course, if regularization is so large that the model cannot fit low-frequency classes, the loss per class after training reflects the class frequencies and the gap between GD and Adam is lessened. ($L_2$ penalty impacts performance on low-frequency more. Take a simple example with one class corresponding to $50\%$ of the data and $100$ classes corresponding to $.5\%$ of the data (each). Increasing $w_1$ can decrease the loss on ${\approx}50\%$ of the data. To achieve the same reduction on the remaining ${\approx}50\%$ of samples requires increasing $w_2, ..., w_{100}$, which is much more costly in $L_2$ penalty)
>
> Those examples illustrate that while the observed behaviour does require the model to be expressive enough so that the model can fit low-frequency classes (otherwise, not fitting low-frequency classes will not be a problem), but it does not require separability nor that the weights diverge to $\infty$.
>
> If the question regarding separability was aimed at the theoretical analysis, then we of course agree that the toy model of §3.3 relies on separability to obtain the result in Theorem 3. We believe that the conclusion that sign or per-class normalized methods can outperform GD holds more broadly. But finding a reasonable model where the computations can be carried out in closed form is far from trivial. We do not think that the gradient flow equations have a closed-form if we add an $L_2$ regularization term, and an analysis for discrete time gradient descent is even more complex.

---

> > ### Comment · Reviewer_HWpb · 2024-08-12
> >
> > Thank you for the response. I will keep my score.

---

### Official Review · Reviewer_ujWn · 2024-07-13

**Soundness:** 4
**Presentation:** 4
**Contribution:** 3
**Rating:** 7
**Confidence:** 4

**Summary:**

This paper investigates the reason why Adam outperforms (S)GD by a large margin on language models when the performance gap is much smaller in other settings. The authors argue that language data often has a heavy-tailed class imbalance, where a large fraction of the data consists of several classes with a small number of words, which leads to slow improvement in the training loss of these classes for (S)GD, whereas Adam (or sign descent) is more robust to this type of imbalance. Since the fraction of classes with fewer samples is relatively large, this contributes to a difference in the rate of decrease of the average loss as well, in contrast to conventionally considered settings for binary classification with class imbalance. The authors show that the performance gap can be reproduced in other settings, such as when training CNNs on MNIST, ResNet on ImageNet, and even linear models on synthetic high-dimensional data, when the heavy-tailed class imbalance is introduced in these datasets. The authors also show that similar effects are observed without stochasticity and with a simpler algorithm (sign gradient descent), and present some theoretical results in simple settings showing a difference in the rate of decrease in the training loss between GD and sign GD.

**Strengths:**

- The paper presentation is exceptional: it is very well-written with aesthetic plots that illustrate the results really clearly. The contribution, as well as the motivation for each of the experiments conducted in the paper, is discussed in detail; I especially like the last paragraph of Section 1.1.

- The experiments are designed well and thoroughly support the hypothesis that heavy-tailed class imbalance is a key reason for the performance gap between SGD and Adam on language models. Results showing that the gap can be reproduced in other settings when the heavy-tailed class imbalance is introduced are very convincing.

- The paper contributes to our understanding of when and why Adam performs better than GD, which is fundamental to ultimately improving optimization.

**Weaknesses:**

There are no major weaknesses. The authors adequately acknowledge and discuss the limitations of the work. Some minor concerns are as follows.

- The theoretical results are in oversimplified settings. However, this seems justified since theory is not one of the primary contributions of the paper. It might be better to allocate less space to this part (e.g., Section 3.3).

- It would be nice to elaborate on lines 331-333 and include more discussion on this aspect in Section 3.

- Some corrections/suggestions:

    - Missing ‘the’ in line 152. Extra ‘the’ in line 225. Missing ‘than’ in line 607. Extra ‘be’ in line 631.

    - It would be good to add some description in lines 496-497.

**Questions:**

For the Barcoded MNIST dataset, I think the number of new images should be $5\times 10\times (2^{10}-1)$ since one of the 10-bit patterns would be the same as the background. Can the authors clarify this?

**Limitations:**

There are no potential negative impacts of this work. The authors discuss the limitations of their work at length in Section 4.

---

> ### Author Rebuttal · Authors · 2024-08-05
>
> **Clarification on barcoded MNIST**
> > For the Barcoded MNIST dataset, I think the number of new images should be $5\times 10 \times (2^{10}-1)$ since one of the 10-bit patterns would be the same as the background. Can the authors clarify this?
>
> Good catch, thanks! You are correct, the all-0 10-bit pattern would be indistinguishable from the background. This was a typo in the appendix, which should read "$10 \times (2^{10}-1)$ classes". The code did generate $2^{10}-1$ barcodes, excluding the all-0 string (`generate_combinations` in `code/src/optexp/datasets/barcoded_mnist.py:l27-33`). We will fix the appendix.

---

> > ### Comment · Reviewer_ujWn · 2024-08-11
> >
> > Thank you for the rebuttal and the clarification, I will maintain my score.

---

### Author Rebuttal · Authors · 2024-08-05

We thank the reviewers for their careful readings and thoughtful comments.

We answer specific questions in individual replies:
- [**ujWn:** Clarification on barcoded MNIST](https://openreview.net/forum?id=T56j6aV8Oc&noteId=2QGkcNTqaR)
- [**HWpb:** Does the conclusion still hold when the parameters are not separable](https://openreview.net/forum?id=T56j6aV8Oc&noteId=ASSfu6V1bw)
- [**LhRX:** Does GD get stuck on Imbalanced MNIST (Fig. 2)](https://openreview.net/forum?id=T56j6aV8Oc&noteId=cKpTYeF9u4)
- [**CdGj:** Impact of class imbalance beyond the last layer](https://openreview.net/forum?id=T56j6aV8Oc&noteId=vrl5iFlM2d)

We will address the minor comments such as typos in a revision.

---

### Decision · Program_Chairs · 2024-09-25

**Decision:**

Accept (spotlight)

**Comment:**

This paper provides new insights into why Adam consistently outperforms SGD by a significant margin in language models. While this phenomenon has been observed by both practitioners and researchers for some time, previous explanations have largely focused on the architectures of models like Transformers. In contrast, this paper introduces a new hypothesis, proposing that the heavy-tailed class imbalance in training datasets is the key factor driving this performance gap. The authors conducted a series of rigorous experiments that solidly support their hypothesis, demonstrating that this gap is reproducible across various tasks and architectures. Additionally, the paper offers theoretical analysis in simplified settings, which deepens our fundamental understanding of this phenomenon.

All reviewers unanimously agreed on the acceptance and gave high ratings. I recommend accepting it as a spotlight paper, given its solid foundation and the potential for significant impact on the field.